# OFFLINE REINFORCEMENT LEARNING WITH GENERATIVE TRAJECTORY POLICIES

## ABSTRACT

Generative models have emerged as a powerful class of policies for offline reinforcement learning (RL) due to their ability to capture complex, multi-modal behaviors. However, existing methods face a stark trade-off: slow, iterative models like diffusion policies are computationally expensive, while fast, single-step models like consistency policies often suffer from degraded performance. In this paper, we demonstrate that it is possible to bridge this gap. The key to moving beyond the limitations of individual methods, we argue, lies in a unifying perspective that views modern generative models—including diffusion, flow matching, and consistency models—as specific instances of learning a continuous-time generative trajectory governed by an Ordinary Differential Equation (ODE). This principled foundation provides a clearer design space for generative policies in RL and allows us to propose *Generative Trajectory Policies* (GTPs), a new and more general policy paradigm that learns the entire solution map of the underlying ODE. To make this paradigm practical for offline RL, we further introduce two key theoretically principled adaptations. Empirical results demonstrate that GTP achieves state-of-the-art performance on D4RL benchmarks – it significantly outperforms prior generative policies, achieving perfect scores on several notoriously hard AntMaze tasks.

## 1 INTRODUCTION

In offline reinforcement learning (RL), an agent needs to learn a policy from a pre-collected dataset without any further interaction with the environment. This setting creates a fundamental challenge: the agent is asked to generalize from limited, often narrow, experience to an unpredictable world. At the heart of this challenge lies the need for policy expressiveness - the capacity to capture rich, often multi-modal patterns of behavior present in real-world datasets. Traditional offline RL methods rely on simple function approximators and are prone to distribution shift, as the learned policy may choose actions not present in the dataset, leading to inaccurate value estimates (Fujimoto et al., 2019; Wu et al., 2019; Kumar et al., 2020). This has sparked growing interest in generative models - from generative adversarial networks (GANs), variational autoencoders (VAEs), to Energy-Based Models (EBMs) - as powerful tools to model the full complexity and diversity of RL policies (Ho & Ermon, 2016; Ha & Schmidhuber, 2018; Ho et al., 2020; Brahmanage et al., 2023; Messaoud et al., 2024).

Most recently, diffusion-based policies have emerged as a powerful paradigm due to their exceptional ability to represent complex, multi-modal distributions (Wang et al., 2023; Janner et al., 2022; Pearce et al., 2023). However, their expressive power comes at a steep price: the slow, iterative sampling process required for generation imposes a significant computational burden, hindering their practical utility. To resolve this, subsequent work has employed consistency-based models to accelerate inference, often enabling one or two-step generation (Ding & Jin, 2024). While remarkably fast, this simplification frequently leads to degraded policy quality, with performance saturating quickly.

This demonstrates a fundamental trade-off between expressiveness and efficiency for generative policies. The research question in this work is: *Is it possible to design a policy class that can achieve both policy expressiveness and computational efficiency?*

A key insight of our work is that the path to resolving this trade-off lies in a general principle that unifies a family of powerful modern generative models. We observe that a spectrum of recent advancements, including diffusion models (Song & Ermon, 2019; Song et al., 2021b), Consistency Models (Song et al., 2023), Consistency Trajectory Models (CTMs) (Kim et al., 2024), and various

forms of Flow Matching (Frans et al., 2025; Geng et al., 2025), can all be understood through the lens of a continuous-time generative trajectory governed by an Ordinary Differential Equation (ODE). This unified perspective provides the theoretical foundation for our work, enabling us to conceptualize a policy itself as a full trajectory and thereby design a new class of expressive and efficient policies.

Building on this foundation, we introduce *Generative Trajectory Policies* (GTPs), a new policy paradigm that learns the entire solution map of the underlying ODE. By learning the full trajectory, GTPs are not confined to either slow, high-fidelity sampling or fast, low-fidelity shortcuts. Instead, they enable flexible, multi-step, deterministic generation that can achieve high performance even with a few sampling steps.

Our key contributions include: i) We propose GTP, a new and highly expressive policy paradigm for offline RL, derived from a unifying framework that connects a family of modern generative models to continuous-time ODE trajectories. ii) We make a practical implementation of the GTP paradigm by developing two key *theoretically-grounded adaptations* that address computational cost, training instability, and misaligned objectives, including a score approximation and a variational framework for value-driven policy improvement. iii) We empirically validate GTP on the D4RL benchmarks, where it achieves state-of-the-art performance, outperforming prior generative and offline RL methods. Notably, our approach achieves perfect scores on several notoriously challenging AntMaze tasks, demonstrating its ability to strike a more favorable balance between expressiveness and efficiency. Our code is included in the supplementary and will be released upon paper acceptance.

## 2 RELATED WORK

We briefly introduce the related work. A more detailed discussion is provided in Appendix A.

**Expressive Policies in Offline RL.** Offline RL depends on policies that are expressive enough to capture the diverse, often multi-modal behaviors present in datasets. Conventional choices like Gaussian policies are easy to train but struggle to represent such complexity. Much of the literature has instead advanced from the critic side, regularizing value functions to guard against overestimation (Fujimoto et al., 2019; Wu et al., 2019; Kumar et al., 2020; Kostrikov et al., 2022). While effective, these methods leave the policy class itself underpowered, motivating a complementary line of research: actor-centric approaches that adopt generative models. Early explorations with GANs/VAEs (Ho & Ermon, 2016) and energy-based policies (Messaoud et al., 2024) showed promise, but were often hampered by training instabilities and did not achieve the sample quality of modern generative paradigms, leaving the need for a truly robust and expressive policy class as a key open problem.

**Continuous-Time Generative Models.** A new generation of powerful tools for this task has emerged from the generative modeling community. A spectrum of recent advancements can be understood through the unifying lens of learning a continuous-time trajectory governed by an Ordinary Differential Equation (ODE). This includes score-based diffusion models (Song et al., 2021b), Flow Matching (FM) (Lipman et al., 2023; Frans et al., 2025), Consistency Models (CMs) (Song et al., 2023) and Consistency Trajectory Models (CTMs) (Kim et al., 2024). While this evolution has produced a powerful toolbox of trajectory-based generative models, their potential has not yet been fully realized in the RL domain. The challenge of adapting these powerful but complex models to the specific constraints and objectives of offline RL remains a significant barrier.

**The Trade-off in Generative Policies.** Researchers have begun to apply these powerful generative tools as policies in offline RL. Early work with diffusion-based policies demonstrated their immense potential to model complex action distributions (Wang et al., 2023; Janner et al., 2022; Pearce et al., 2023), but at the cost of slow, iterative inference. In response, consistency-based policies were introduced to accelerate sampling, often to one or two steps (Ding & Jin, 2024), but this frequently resulted in degraded policy performance. This work has established a new and critical trade-off between expressiveness and efficiency. How to properly adapt the underlying principles of these powerful generative models to create a policy class that is both high-performing and efficient in the demanding offline RL setting remains a key open problem that our work aims to address.

## 3 A UNIFIED ODE FRAMEWORK FOR GENERATIVE MODELS

A cornerstone of many modern generative models is the idea of reversing a process that gradually perturbs data into noise. While prior work typically treats diffusion models, consistency models, and

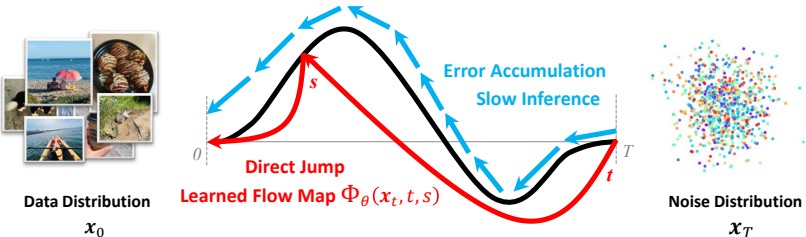

Figure 1: Illustration of the unified solution map $\Phi(\boldsymbol{x}_t, t, s)$. Iterative solvers (blue) suffer from slow inference and error accumulation, whereas the learned flow map (red) enables direct jumps from the noise distribution $x_T$ to the data distribution $x_0$, providing an intuitive view of our unified framework.

flow matching as separate families, we propose a single unified ODE framework that reveals all these models as instances of the same underlying formulation. The reverse process can be described by a general ODE:

$$\frac{\mathrm{d}\boldsymbol{x}_t}{\mathrm{d}t} = f(\boldsymbol{x}_t, t), \tag{1}$$

where the vector field $f(\boldsymbol{x}, t)$ defines a deterministic trajectory from a point $\boldsymbol{x}_T$ sampled from a simple prior distribution to a data sample $\boldsymbol{x}_0$, and $t \in [0, T]$.

Innovation within this framework has advanced along two complementary axes: (1) defining the vector field $f$, as in diffusion-based approaches (Song et al., 2021b) and flow matching (Lipman et al., 2023); and (2) solving the ODE efficiently, which is a central challenge since standard numerical solvers require hundreds of discretization steps, leading to slow inference and accumulated errors (Song et al., 2023; Kim et al., 2024). While the first axis has largely matured, the second remains a bottleneck and has inspired a new class of methods that directly learn the ODE's solution map.

## 3.1 Defining the Vector Field

The choice of the vector field $f(\boldsymbol{x}_t, t)$ is crucial, as it determines the exact generative path from noise to data. Prominent methods for defining these dynamics include:

**Diffusion Models.** Originating from diffusion-based modeling, this class of methods defines the vector field indirectly. The dynamics are determined by the score function, $\nabla_{\boldsymbol{x}_t} \log p_t(\boldsymbol{x}_t)$, which is the gradient of the log-density of the noisy data distribution. In the corresponding Probability Flow (PF) ODE (Song et al., 2021b), a neural network is trained to approximate this score (or an equivalent denoising function), thereby implicitly specifying the vector field that governs the generative process.

**Flow Matching.** In contrast, Flow Matching (FM) (Lipman et al., 2023) provides a more direct and general framework for learning the vector field. This method involves training a neural network $f_{\boldsymbol{\theta}}(\boldsymbol{x}_t, t)$ by directly regressing it against a known target vector field that connects the data and prior distributions. This direct regression offers a stable and often more efficient training objective.

## 3.2 Efficiently Solving the ODE by Learning the Solution Map

Instead of relying on numerical integration, a powerful alternative is to model the ODE's solution map directly. We highlight that the true ODE flow map, $\Phi(\boldsymbol{x}_t, t, s)$, which maps a state at time $t$ to its corresponding state at time $s$, naturally provides a *unifying representation* for a wide family of generative models:

$$\boldsymbol{x}_s = \Phi(\boldsymbol{x}_t, t, s) = \boldsymbol{x}_t + \int_t^s f(\boldsymbol{x}_\tau, \tau)\mathrm{d}\tau. \tag{2}$$

Figure 1 illustrates this unified viewpoint. Under this formulation, classic approaches such as Consistency Models (Song et al., 2023), Consistency Trajectory Models (Kim et al., 2024), Shortcut Models (Frans et al., 2025), and Mean Flows (Geng et al., 2025) can all be interpreted as approximating specific aspects or limits of the same flow map $\Phi$. For instance, diffusion denoisers estimate its infinitesimal form, whereas consistency models enforce its compositional structure.

### 3.3 LEARNING THE FLOW MAP: A GENERAL PARAMETERIZATION

Guided by this unified perspective, we introduce a general parameterization for learning the ODE flow map. Although $\Phi$ provides the ideal target, its integral form is not directly suitable for training. We therefore adopt a surrogate function $\phi(\boldsymbol{x}_t, t, s)$ inspired by (Kim et al., 2024):

$$\phi(\boldsymbol{x}_t, t, s) = \boldsymbol{x}_t + \frac{t}{t-s} \int_t^s f(\boldsymbol{x}_\tau, \tau) \mathrm{d}\tau. \tag{3}$$

The exact flow map can be recovered via linear interpolation:

$$\Phi(\boldsymbol{x}_t, t, s) = \left(1 - \frac{s}{t}\right) \phi(\boldsymbol{x}_t, t, s) + \frac{s}{t} \boldsymbol{x}_t. \tag{4}$$

This parameterization has a natural interpretation: $\phi(\boldsymbol{x}_t, t, s)$ serves as an estimate of the endpoint $\boldsymbol{x}_0$, extrapolated from $\boldsymbol{x}_t$ using the *average velocity* over $[s, t]$. Importantly, it allows us to define two complementary training objectives that together form the core of our unified ODE framework.

**1. The Instantaneous Flow Loss (Local Anchor).** This objective ensures the learned map is correct for *infinitesimal steps* by enforcing a boundary condition at the limit $s \to t$:

$$\lim_{s \to t} \phi(\boldsymbol{x}_t, t, s) = \boldsymbol{x}_t - t f(\boldsymbol{x}_t, t). \tag{5}$$

For convenience, we denote $\phi^{\text{inst}}(\boldsymbol{x}_t, t) := \phi(\boldsymbol{x}_t, t, t)$ and refer to it as the *Inst Map*. This condition provides a powerful connection to prominent generative modeling paradigms: the right-hand side recovers the denoiser $D(\boldsymbol{x}_t, t)$ (i.e., $\mathbb{E}[\boldsymbol{x}_0 \mid \boldsymbol{x}_t]$) in diffusion models and the velocity field target in flow matching, $f(\boldsymbol{x}_t, t) = (\boldsymbol{x}_t - \phi^{\text{inst}}(\boldsymbol{x}_t, t))/t$. In practice, $\phi_{\boldsymbol{\theta}}^{\text{inst}}(\boldsymbol{x}_t, t)$ is the model prediction, trained with task-specific targets: for diffusion, the target is the clean sample $\boldsymbol{x}_0$; for flow matching, the target becomes $\boldsymbol{x}_t - t(\boldsymbol{x}_1 - \boldsymbol{x}_0)$. In this sense, the Inst Map acts as a *local anchor*, unifying diffusion-style denoising and flow-matching velocity estimation under a single principle.

**2. The Trajectory Consistency Loss (Global Regulator).** This objective enforces correctness across long, *multi-step jumps* by requiring self-consistency:

$$\Phi(\boldsymbol{x}_t, t, s) \approx \Phi(\Phi(\boldsymbol{x}_t, t, u), u, s), \quad \text{for } t > u > s. \tag{6}$$

where $u$ denotes an intermediate time between $t$ and $s$. Here, the displacement over $[t, s]$ must equal the sum of displacements over $[t, u]$ and $[u, s]$. In practice, the right-hand side is treated as the target: $\Phi(\boldsymbol{x}_t, t, u)$ is obtained using an ODE solver (or its learned approximation), and then composed forward to $s$. The loss is then defined by the discrepancy between the left- and right-hand sides of Eq. (6). This serves as a *global regulator*, enforcing coherence of long trajectories with the additive structure of ODEs.

Taken together, the two objectives are complementary: the instantaneous loss enforces fidelity in local dynamics, while the trajectory consistency loss guarantees global coherence across time. We next show how several prominent generative models emerge as a concrete instances of this unified ODE framework.

### 3.4 PRIOR MODELS AS SPECIAL CASES OF THE UNIFIED ODE FRAMEWORK

With the flow map $\Phi$, the reparameterization $\phi$, and the two training objectives, many existing generative models naturally emerge as a special case of our unified ODE framework. Below we summarize the key correspondences; additional details are provided in Appendix B.1

**Consistency Models (CMs).** CMs (Song et al., 2023) effectively restrict the flow map to the terminal-time evaluation $\Phi(\boldsymbol{x}_t, t, 0)$ in Eq. (2). Their core objective enforces a discrete approximation of the ODE's compositional property:

$$\Phi(\boldsymbol{x}_t, t, 0) \approx \Phi(\Phi(\boldsymbol{x}_t, t, t - \Delta t), t - \Delta t, 0),$$

which is directly aligned with our Trajectory Consistency Loss in Eq. (6). The EMA bootstrapping operator is a practical implementation of enforcing the same flow-map identity with a one-step target.

**Consistency Trajectory Models (CTMs).** CTMs (Kim et al., 2024) explicitly parameterize $\Phi(\boldsymbol{x}_t, t, s)$ and train it using a form of trajectory self-consistency. This corresponds exactly to our

Trajectory Consistency Loss, while their auxiliary diffusion loss plays the role of our Instantaneous Flow Loss. Thus CTMs instantiate both core components of our unified framework.

**Shortcut Models.** Shortcut Models (Frans et al., 2025) learn a finite-time "average velocity" over the interval $[t, t + d]$, which corresponds to estimating the integral term in the reparameterized form $\phi$ of Eq. (3). Taking the limit $d \to 0$ yields the instantaneous velocity $f(\boldsymbol{x}_t, t)$ and therefore aligns with Eq. (5), while imposing compatibility across different $d$ values realizes a discrete form of the Trajectory Consistency condition in Eq. (6). Hence Shortcut Models can be viewed as learning finite-time approximations to the same flow map $\Phi$.

**Mean Flows.** Mean Flows (Geng et al., 2025) reparameterize the average velocity

$$u(\boldsymbol{x}_t, t, s) = \frac{\boldsymbol{x}_t - \phi(\boldsymbol{x}_t, t, s)}{t},$$

which follows directly from the flow-map definition in Eq. (2) and aligns with our $\phi$ parameterization in Eq. (3). Instead of applying an explicit multi-step consistency loss, Mean Flows enforce a differential "MeanFlow Identity" relating $u$ and the instantaneous velocity $f$, serving as an implicit analog of the Trajectory Consistency condition in Eq. (6). Despite using a distinct training mechanism, the learned object is mathematically a special case of our reparameterized flow representation.

## 4 GENERATIVE TRAJECTORY POLICIES FOR OFFLINE RL

In the previous section, we established a unified ODE trajectory framework that offers an elegant lens for understanding a family of modern generative models. This lays a theoretical foundation for designing expressive generative trajectory policies. We define a Generative Trajectory Policy (GTP) as a policy class that generates actions by learning the solution map of a continuous-time generative ODE. However, translating these insights into a functional offline RL algorithm is hindered by three practical challenges:

**Prohibitive Computational Burden.** Learning an ODE trajectory requires on-trajectory supervision. As discussed in Section 3.3, this is obtained by numerically solving the ODE backward from $t$ to an intermediate point $u$ using multiple discrete steps (e.g., Euler, Heun), an operation we denote as $\mathrm{Solver}(\boldsymbol{x}_t, t, u)$. When scaled to offline RL, where millions of updates are needed, repeatedly performing this inner-loop solving for every sample makes the overall computation quickly intractable.

**Inherent Training Instability.** Unlike distillation methods, our framework must learn the entire ODE trajectory *from scratch*. Central to this process is the Inst Map $\phi^{\mathrm{inst}}(\boldsymbol{x}_t, t)$, which specifies the ODE's right-hand side through $f(\boldsymbol{x}_t, t) = (\boldsymbol{x}_t - \phi^{\mathrm{inst}}(\boldsymbol{x}_t, t))/t$. Early in training, the Inst Map is highly inaccurate; yet its outputs are immediately fed back into the solver to generate supervision. This bootstrapping quickly forms a vicious cycle that resembles TD learning (Sutton, 1988)—bad targets yield bad updates—that destabilizes the Actor–Critic loop and often hinders convergence.

**Misaligned Generative Objective.** The default objective of generative models is to match the data distribution, which in offline RL reduces to behavior cloning (BC). While BC is a reasonable baseline, it cannot achieve policy improvement—the central goal of offline RL. Thus, a key challenge is to design a value-aware objective that leverages the generative process not only to imitate observed actions but also to emphasize those leading to higher returns.

To address these challenges, we introduce two key techniques tailored to the practical implementation of GTP, as illustrated in Figure 2. The following subsections detail these techniques and show how they jointly enable stable, efficient, and value-driven training.

### 4.1 EFFICIENT AND STABLE TRAINING VIA SCORE APPROXIMATION

A central difficulty in our framework is the reliance on self-referential supervision: the model must repeatedly supply $\phi^{\mathrm{inst}}(\boldsymbol{x}_t, t)$ (score estimates) at each solver time point[1], which the ODE solver integrates over many iterations. This approach is not only computationally demanding, but also fragile—early-stage errors in the learned vector field immediately corrupt the supervision signals.

---

[1]Throughout the paper we use the term *score* for consistency with prior literature, although in our framework it is formally the Inst Map $\phi^{\mathrm{inst}}$.

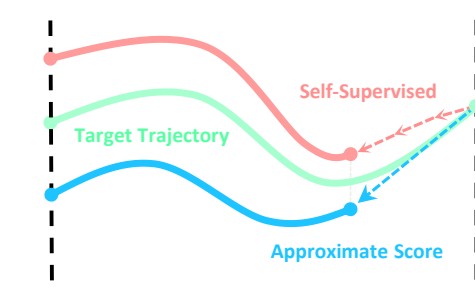 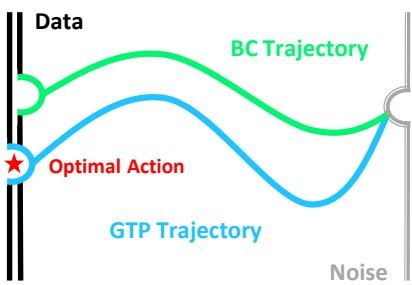

(a) Stable Score Approximation       (b) Value-Driven Guidance

Figure 2: The two core techniques of the GTP implementation: (a) Stable Score Approximation: the target trajectory (green) is contrasted with a reference (red) computed by a multi-step ODE solver (red dashed arrow). The blue dashed arrow denotes a single-step update obtained from our approximate score, which yields the blue trajectory without multi-step integration. (b) Value-Driven Guidance: the BC trajectory (green) is shifted toward high-value regions so that the learned GTP trajectory (blue) approaches the optimal action while remaining aligned with the data.

To address it, we replace $\phi^{\text{inst}}(\boldsymbol{x}_t, t)$ with a closed-form surrogate anchored to the offline sample, $\tilde{f}(\boldsymbol{x}_t, t) = (\boldsymbol{x}_t - \boldsymbol{x})/t$. The theorem below shows that this yields a training loss asymptotically equivalent to the ideal one.

**Theorem 1.** *Fix a time horizon $T > 0$, let $\boldsymbol{x} \sim p_{\text{data}}$, $\boldsymbol{z} \sim \mathcal{N}(0, I)$, and define $\boldsymbol{x}_t = \boldsymbol{x} + t\boldsymbol{z}$. Define the vector fields $f^\star, \tilde{f} : \mathbb{R}^d \times (0, T] \to \mathbb{R}^d$ by $f^\star(\boldsymbol{x}_t, t) := \frac{\boldsymbol{x}_t - \mathbb{E}[\boldsymbol{x}|\boldsymbol{x}_t]}{t}$ and $\tilde{f}(\boldsymbol{x}_t, t) := \frac{\boldsymbol{x}_t - \boldsymbol{x}}{t}$. Assume $f^\star(\cdot, t)$ is Lipschitz in $x$. Let $t = \tau_0 > \tau_1 > \cdots > \tau_K = u$ be a sequence of time points with step sizes $\Delta_k = \tau_{k+1} - \tau_k$ and maximal step $h = \max_k |\Delta_k|$. For a $p$-th order, zero-stable one-step solver $S_{\Delta_k}[f] : \mathbb{R}^d \to \mathbb{R}^d$, define the multi-step propagation from $t$ to $u$ as*

$$\Psi^{\text{sol}}_{t \to u}[f] := S_{\Delta_{K-1}}[f] \circ \cdots \circ S_{\Delta_0}[f]. \tag{7}$$

*Assume further that for each $u > s$, $\Phi_{\boldsymbol{\theta}}(\cdot, u, s) : \mathbb{R}^d \to \mathbb{R}^d$ is Lipschitz in $x$, and that solver states admit bounded second moments independent of $h$. Define the **ideal** and **practical** training objectives*

$$\mathcal{L}_{\text{ideal}}(\boldsymbol{\theta}) := \mathbb{E}\Big[\big\|\Phi_{\boldsymbol{\theta}}(\boldsymbol{x}_t, t, s) - \Phi_{\boldsymbol{\theta}^-}\big(\Psi^{\text{sol}}_{t \to u}[f^\star](\boldsymbol{x}_t), u, s\big)\big\|^2\Big], \tag{8}$$

$$\mathcal{L}_{\text{prac}}(\boldsymbol{\theta}) := \mathbb{E}\Big[\big\|\Phi_{\boldsymbol{\theta}}(\boldsymbol{x}_t, t, s) - \Phi_{\boldsymbol{\theta}^-}\big(\Psi^{\text{sol}}_{t \to u}[\tilde{f}](\boldsymbol{x}_t), u, s\big)\big\|^2\Big], \tag{9}$$

*where $\Phi_{\boldsymbol{\theta}^-}$ denotes the exponentially moving averaged model. Then*

$$\big|\mathcal{L}_{\text{prac}}(\boldsymbol{\theta}) - \mathcal{L}_{\text{ideal}}(\boldsymbol{\theta})\big| = O(h^p). \tag{10}$$

*In particular, as $h \to 0$, the two objectives coincide in expectation.*

*Proof Sketch.* The only difference between the two objectives is that the solver uses the surrogate $\tilde{f}$ instead of the true field $f^\star$. Since both are Lipschitz and the solver is $p$-th order and zero-stable, the propagated states differ by $O(h^p)$ in mean square. By Lipschitz continuity of $\Phi_{\boldsymbol{\theta}}$, this discrepancy transfers directly to the objectives, giving the stated bound. Details are deferred to Appendix B.3. □

Theorem 1 shows that using the closed-form surrogate $\tilde{f}$ changes the objective only by $O(h^p)$, providing theoretical support for our formulation. This replacement makes GTP training both efficient and robust in offline RL, which is further validated empirically by our ablation study (Section 5.3). Further intuition is given in Appendix B.4, where we relate this formulation to consistency training and flow matching.

**Remark 1** (Computational Efficiency). *Using the surrogate score removes the need for multi-step ODE integration. Intermediate points $\boldsymbol{x}_u$ for the trajectory consistency loss are obtained directly as*

$$\boldsymbol{x}_u = \boldsymbol{x} + u \cdot \boldsymbol{z}, \quad \boldsymbol{z} \sim \mathcal{N}(0, I), \tag{11}$$

*a one-step perturbation instead of a costly numerical solver.*

**Remark 2** (Training Stability). *Anchoring supervision to offline data avoids the instability of self-generated targets. The model no longer relies on imperfect early-stage estimates of its own vector field, but instead receives a stable analytical signal tied directly to $x$. This breaks the cycle of error propagation and ensures consistent learning from the very beginning of training.*

## 4.2 VALUE-DRIVEN GUIDANCE FOR POLICY IMPROVEMENT

To address the misaligned generative objective and unify generative imitation with value-based policy improvement, we formalize a value-weighted training objective for our GTP in Theorem 2, with the detailed derivation provided in Appendix B.5.

**Theorem 2** (Advantage-Weighted Objective). *Consider the KL-regularized policy optimization problem in offline RL. Its optimal solution can be written as*

$$\pi^*(a|s) \propto \pi_{\mathrm{BC}}(a|s) \exp\big(\eta A(s,a)\big), \tag{12}$$

*where $A(s,a) = Q(s,a) - V(s)$ is the advantage. Training a generative policy $\pi_{\boldsymbol{\theta}}$ to match $\pi^*$ is therefore equivalent to solving the weighted generative training objective*

$$\max_{\boldsymbol{\theta}} \; \mathbb{E}_{(s,a)\sim\mathcal{D}}\big[\exp\big(\eta A(s,a)\big)\,\ell_{gen}(\pi_{\boldsymbol{\theta}};a|s)\big], \tag{13}$$

*where $\ell_{gen}$ denotes the standard generative loss (e.g., diffusion loss, or flow-matching loss).*

Theorem 2 confirms that exponential advantage weighting is the theoretically correct way to incorporate value guidance into generative training.

**Remark 3** (Practical Implementation). *For numerical stability, we normalize the advantage weights and truncate negatives:*

$$w(s,a) = \exp\bigg(\eta \cdot \frac{\max(0, A(s,a))}{\mathrm{std}(A) + \epsilon}\bigg). \tag{14}$$

*This ensures stable optimization while allowing GTP to preferentially imitate high-advantage actions, thereby preserving the robustness of standard generative training.*

## 4.3 THE GTP OPTIMIZATION FRAMEWORK

Having introduced the two key techniques for the practical implementation of our GTP paradigm, we now integrate them into a complete actor-critic algorithm. The actor is our Generative Trajectory Policy, $\pi_{\boldsymbol{\theta}}$, represented by the learned solution map $\Phi_{\boldsymbol{\theta}}$. The critic is a standard double Q-network, $Q_{\boldsymbol{\varphi}}$, trained to estimate state-action values.

**Policy Representation and Action Sampling.** The actor $\Phi_{\boldsymbol{\theta}}(s, a_t, t, \tau)$ learns to map a noisy action $a_t$ at time $t$ to a cleaner action $a_\tau$ at time $\tau \leq t$, conditioned on a state $s$. At inference time, an action is generated by starting with pure Gaussian noise $a_T \sim \mathcal{N}(0, T^2\mathbf{I})$ and iteratively applying the learned map over a sequence of timesteps $T = t_0 > t_1 > ... > t_K = 0$:

$$a_{t_{i+1}} = \Phi_{\boldsymbol{\theta}}(s, a_{t_i}, t_i, t_{i+1}), \quad \text{for } i = 0, \ldots, K-1. \tag{15}$$

The final denoised sample $a_0$ is the action executed by the policy, i.e., $\pi_{\boldsymbol{\theta}}(s) := a_0$.

**Critic Training.** We use a standard double Q-network parameterized by $\phi$ to mitigate the overestimation bias. The critic is trained to minimize the temporal-difference (TD) error using a batch of transitions $(s, a, r, s')$ from the offline dataset:

$$\mathcal{L}_{\mathrm{critic}} = \mathbb{E}\bigg[\bigg(r + \gamma \cdot \min_{j=1,2} Q_{\boldsymbol{\varphi}_j^-}(s', \pi_{\boldsymbol{\theta}'}(s')) - Q_{\boldsymbol{\varphi}_j}(s,a)\bigg)^2\bigg] \tag{16}$$

where $\boldsymbol{\varphi}^-$ and $\boldsymbol{\theta}^-$ are the target networks for the critic and actor, respectively, updated via exponential moving average (EMA).

**Actor Training.** The GTP actor is trained by combining the two fundamental objectives in Eqs.(5)-(6) introduced in our unified framework. These objectives are directly modified to incorporate our key adaptations for offline RL. To enable policy improvement, both loss components are weighted by the advantage-based term $w(s,a)$, thereby prioritizing high-value actions. Simultaneously, to

ensure computational feasibility and training stability, the supervision targets are generated using our efficient score approximation instead of a costly ODE solver. First, the Trajectory Consistency Loss, $\mathcal{L}_{\text{Consistency}}$, enforces the global self-consistency of the learned flow map $\Phi_{\boldsymbol{\theta}}$:

$$\mathcal{L}_{\text{Consistency}} = \mathbb{E}_{(s,a)\sim\mathcal{D}} \, \mathbb{E}_{t,\tau,u} \, \mathbb{E}_{\boldsymbol{z}\sim\mathcal{N}(0,I)} \Big[ w(s,a) \, \|\Phi_{\boldsymbol{\theta}}(s,a_t,t,\tau) - \Phi_{\boldsymbol{\theta}^-}(s,\tilde{a}_u,u,\tau)\|_2^2 \Big]. \quad (17)$$

where $a_t = a + t \cdot z$, and the teacher's intermediate action $\tilde{a}_u = a + u \cdot z$. Second, the Instantaneous Flow Loss, $\mathcal{L}_{\text{Flow}}$, anchors the model's local dynamics. As established in Section 3.3, this objective enforces that the learned Inst Map behaves as a correct denoiser in the infinitesimal limit. We implement it by penalizing the prediction error of $\phi_{\boldsymbol{\theta}}^{\text{inst}}$:

$$\mathcal{L}_{\text{Flow}} = \mathbb{E}_{(s,a)\sim\mathcal{D}} \, \mathbb{E}_t \Big[ w(s,a) \, \| a - \phi_{\boldsymbol{\theta}}^{\text{inst}}(s,a_t,t)\|_2^2 \Big]. \quad (18)$$

The total actor loss is then a weighted sum of the two components:

$$\mathcal{L}_{\text{actor}} = \mathcal{L}_{\text{Consistency}} + \lambda_{\text{Flow}} \cdot \mathcal{L}_{\text{Flow}}. \quad (19)$$

The full training pipeline is outlined in Algorithm 1.

---

**Algorithm 1** Training Generative Trajectory Policy (GTP)

---

1: Initialize actor $\Phi_{\boldsymbol{\theta}}$, critic $Q_{\boldsymbol{\varphi}}$, target networks $\boldsymbol{\theta}^- \leftarrow \boldsymbol{\theta}$, $\boldsymbol{\varphi}^- \leftarrow \boldsymbol{\varphi}$
2: **for** iteration $i = 1$ to $N_{\text{iter}}$ **do**
3:     Sample batch $(s,a,r,s') \sim \mathcal{D}$
4:     Update critic $Q_{\boldsymbol{\varphi}}$ using Eq. (16)
5:     Compute advantage weights $w(s,a)$ using the trained critic
6:     Sample time pairs $t > u > \tau$, and noise $\boldsymbol{z} \sim \mathcal{N}(0,\mathbf{I})$
7:     Generate noisy actions via score approx.: $a_t = a + t \cdot \boldsymbol{z}$, $\tilde{a}_u = a + u \cdot \boldsymbol{z}$
8:     Update actor $\Phi_{\boldsymbol{\theta}}$ using the weighted loss in Eq. (19)
9:     Update target networks: $\boldsymbol{\theta}^- \leftarrow \tau\boldsymbol{\theta} + (1-\tau)\boldsymbol{\theta}^-$,    $\boldsymbol{\varphi}^- \leftarrow \tau\boldsymbol{\varphi} + (1-\tau)\boldsymbol{\varphi}^-$
10: **end for**

---

## 5 EXPERIMENTAL RESULTS

In this section, we empirically validate our central claims through experiments. Our evaluation is designed to answer three core questions: (i) whether GTP provides a more expressive generative model for imitating complex behaviors than prior approaches; (ii) whether our two key techniques (Section 4) effectively translate into stable policy improvement that surpasses state-of-the-art offline RL algorithms; and (iii) whether GTP resolves the tension between expressiveness and efficiency.

We evaluate our method on a suite of challenging offline reinforcement learning tasks from the D4RL benchmark (Fu et al., 2020), including the Gym and AntMaze domains. Following the standard setting of Ding & Jin (2024), we evaluate each policy over 10 episodes for Gym tasks and 100 episodes for all other tasks. Unless otherwise noted, diffusion policies and our GTP use $K = 5$ sampling steps, and consistency policies use $K = 2$. Hyperparameters are provided in Appendix C.1.

Due to space limit, we only show major results in the following. Additional ablations and visualizations in a multi-goal environment are deferred to Appendix D, which provide further evidence of the effectiveness and efficiency of GTP.

### 5.1 EXPRESSIVENESS AS A BEHAVIOR CLONING POLICY

To assess the intrinsic modeling capacity of our policy architecture, we first conduct experiments in a pure behavior cloning (BC) setting. By setting the value-guidance coefficient $\eta = 0$, the objective reduces to a purely generative supervised loss, so the policy is trained only to match the data distribution without policy improvement.

**Baselines.** We compare our method, GTP-BC, against a diverse set of baselines, which includes classic behavior cloning (a Gaussian policy), several strong offline RL methods such as AWAC (Nair et al., 2020) and TD3+BC (Fujimoto & Gu, 2021), and importantly, other generative policies in a BC setting: Diffusion-BC (D-BC) (Wang et al., 2023) and Consistency-BC (C-BC) (Ding & Jin, 2024).

**Results and Analysis.** As shown in Table 1, our method achieves strong results across a broad spectrum of tasks, from basic locomotion to complex sparse-reward environments, ***achieving state-of-the-art performances in 11 out of 15 tasks***. This strong overall performance is reflected in the average scores across both major task suites. In the Gym tasks, our model's average return of 82.3 significantly surpasses both D-BC (76.3) and C-BC (69.7). This highlights the superior modeling capacity of learning the full trajectory map. The performance is even more striking in the notoriously difficult AntMaze suite, where long-horizon planning and multimodality are critical. Here, GTP-BC (66.3) dramatically outperforms all other methods, including the next-best generative approach, C-BC (44.1). This substantial gap suggests that our model's ability to learn the full continuous-time trajectory provides a powerful inductive bias for capturing the complex, temporally extended behaviors required for success. These results confirm the strong expressiveness inherent to the GTP architecture itself.

Table 1: Behavior cloning performances on D4RL. We report the mean and standard deviation of normalized scores over 5 random seeds. Bold indicates the best performance among all methods.

| Gym | BC | AWAC | Diffuser | MoRel | Onestep RL | TD3+BC | DT | **D-BC** | **C-BC** | **GTP-BC (Ours)** |
|---|---|---|---|---|---|---|---|---|---|---|
| halfcheetah-m | 42.6 | 43.5 | 44.2 | **42.1** | 48.4 | 48.3 | 42.6 | 45.4 | 31.0 | **48.6±0.3** |
| hopper-m | 52.9 | 57.0 | 58.5 | **95.4** | 59.6 | 59.3 | 67.6 | 65.3 | 71.7 | 83.7±4.0 |
| walker2d-m | 75.3 | 72.4 | 79.7 | 77.8 | 81.8 | **83.7** | 74.0 | 81.2 | 83.1 | 77.1±1.7 |
| halfcheetah-mr | 36.6 | 40.5 | 42.2 | 40.2 | 38.1 | 44.6 | 36.6 | 41.7 | 34.4 | **46.3±0.6** |
| hopper-mr | 18.1 | 37.2 | 96.8 | 93.6 | 97.5 | 60.9 | 82.7 | 67.3 | 99.7 | **100.5±0.3** |
| walker2d-mr | 26.0 | 27.0 | 61.2 | 49.8 | 49.5 | 81.8 | 66.6 | 77.5 | 73.3 | **83.4±1.8** |
| halfcheetah-me | 55.2 | 42.8 | 79.8 | 53.3 | **93.4** | 90.7 | 86.8 | 90.8 | 32.7 | 91.3±0.5 |
| hopper-me | 52.5 | 55.8 | 107.2 | 108.7 | 103.3 | 98.0 | 107.6 | 107.6 | 90.6 | **109.6±1.9** |
| walker2d-me | 107.5 | 74.5 | 108.4 | 95.6 | **113.0** | 110.1 | 108.1 | 108.9 | 110.4 | 100.2±2.1 |
| **Average** | 51.9 | 50.1 | 75.3 | 72.9 | 76.1 | 75.3 | 74.7 | 76.3 | 69.7 | **82.3** |
| **AntMaze** | BC | AWAC | Diffuser | MoRel | Onestep RL | TD3+BC | DT | **D-BC** | **C-BC** | **GTP-BC (Ours)** |
| antmaze-u | 54.6 | 56.7 | 78.9 | 73.0 | 64.3 | 78.6 | 59.2 | 71.8 | 75.8 | **84.2±6.6** |
| antmaze-ud | 45.6 | 49.3 | 55.0 | 61.0 | 60.7 | 71.4 | 53.0 | 61.2 | 77.6 | **79.2±3.2** |
| antmaze-mp | 0.0 | 0.0 | 0.0 | 0.0 | 0.3 | 10.6 | 0.0 | 43.4 | 56.8 | **74.4±6.5** |
| antmaze-md | 0.0 | 0.7 | 0.0 | 8.0 | 0.0 | 3.0 | 0.0 | 29.8 | 31.6 | **85.0±6.6** |
| antmaze-lp | 0.0 | 0.0 | 6.7 | 0.0 | 0.0 | 0.2 | 0.0 | 14.6 | 10.2 | **34.4±5.1** |
| antmaze-ld | 0.0 | 1.0 | 2.2 | 0.0 | 0.0 | 0.0 | 0.0 | 26.6 | 12.8 | **40.8 ±6.3** |
| **Average** | 16.7 | 18.0 | 23.8 | 23.7 | 20.9 | 27.3 | 18.7 | 41.2 | 44.1 | **66.3** |

## 5.2 From Imitation to Improvement: GTP in Offline RL

Having established GTP's strong performance as an imitation learning agent, we now evaluate the full actor-critic algorithm, GTP, to assess whether our variational policy optimization framework (Section 4.2) can effectively translate this expressiveness into state-of-the-art policy improvement.

**Baselines.** We compare GTP against a suite of strong offline RL algorithms, including CQL (Kumar et al., 2020), IQL (Kostrikov et al., 2021), $\chi$-QL (Garg et al., 2023), ARQ (Goo & Niekum, 2022), IDQL-A (Hansen-Estruch et al., 2023), and the two most relevant generative policy competitors: Diffusion-QL (D-QL) (Wang et al., 2023), QGPO (Lu et al., 2023), BDM (Chen et al., 2024b), and Consistency-AC (C-AC) (Ding & Jin, 2024).

**Results and Analysis.** Table 2 demonstrates that GTP sets a new state-of-the-art for generative policies in offline RL. On the Gym tasks, our method achieves the highest average return (89.0), outperforming the previous best, D-QL (87.9). The gains are even more pronounced in the challenging AntMaze suite, where GTP (80.6) significantly surpasses both Diffusion-QL (69.6) and QGPO (78.3). Notably, on the `antmaze-umaze` task, our method achieves a perfect score of 100.0. These results provide strong evidence that our principled, advantage-weighted learning objective successfully leverages the critic's signal to guide the powerful generative policy beyond simple imitation, enabling robust and effective policy improvement.

## 5.3 Ablation Study

We conduct ablations to evaluate the contribution of two key components of GTP: the score approximation scheme (Section 4.1) and the variational value guidance mechanism (Section 4.2).

**Score Approximation.** Replacing our score approximation with signals generated directly by an ODE solver leads to substantially longer training time and weaker performance, even when the solver is limited to at most three steps. Without approximation, training suffers from high variance and slow convergence due to the need for numerical integration at each iteration. In contrast, our approximation

Table 2: Offline RL results on D4RL (mean ± std over 5 random seeds). Bold indicates best result.

| Gym | CQL | IQL | $\chi$-QL | ARQ | IDQL-A | D-QL | QGPO | BDM | C-AC | GTP (Ours) |
|---|---|---|---|---|---|---|---|---|---|---|
| halfcheetah-m | 44.0 | 47.4 | 48.3 | 45 | 51.0 | 51.1 | 54.1 | 57.0 | **69.1** | 53.9±0.1 |
| hopper-m | 58.5 | 66.3 | 74.2 | 61 | 65.4 | 90.5 | 98.0 | **98.4** | 80.7 | 90.3±2.7 |
| walker2d-m | 72.5 | 78.3 | 84.2 | 81 | 82.5 | 87.0 | 86.0 | 87.4 | 83.1 | **89.5±0.6** |
| halfcheetah-mr | 45.5 | 44.2 | 45.2 | 42 | 45.9 | 47.8 | 47.6 | 51.6 | **58.7** | 50.8±0.4 |
| hopper-mr | 95.0 | 94.7 | 100.7 | 81 | 92.1 | 101.3 | 96.9 | 92.7 | 99.7 | **101.7±0.3** |
| walker2d-mr | 77.2 | 73.9 | 82.2 | 66 | 85.1 | **95.5** | 84.4 | 89.2 | 79.5 | 94.2±0.3 |
| halfcheetah-me | 91.6 | 86.7 | 94.2 | 91 | 95.9 | **96.8** | 93.5 | 93.2 | 84.3 | 93.8±0.8 |
| hopper-me | 105.4 | 91.5 | 111.2 | 110 | 108.6 | 111.1 | 108.0 | 104.9 | 100.4 | **112.2±0.6** |
| walker2d-me | 108.8 | 109.6 | 112.7 | 109 | 112.7 | 110.1 | 110.7 | 111.1 | 110.4 | **114.2±0.3** |
| Average | 77.6 | 77.0 | 83.7 | 76.2 | 82.1 | 87.9 | 86.6 | 87.3 | 85.1 | **89.0** |
| **AntMaze** | **CQL** | **IQL** | **$\chi$-QL** | **ARQ** | **IDQL-A** | **D-QL** | **QGPO** | **BDM** | **C-AC** | **GTP (Ours)** |
| antmaze-u | 74.0 | 87.5 | 93.8 | 97 | 94.0 | 93.4 | 96.4 | 93.0 | 75.8 | **100±0** |
| antmaze-ud | **84.0** | 62.2 | 82.0 | 62 | 80.2 | 66.2 | 74.4 | 81.0 | 77.6 | 81.9±4.4 |
| antmaze-mp | 61.2 | 71.2 | 76.0 | 80 | **84.2** | 76.6 | 83.6 | 79.0 | 56.8 | 83.3±8.1 |
| antmaze-md | 53.7 | 70.0 | 73.6 | 82 | 84.8 | 78.6 | 83.8 | 84.0 | - | **94.2±2.0** |
| antmaze-lp | 15.8 | 39.6 | 46.5 | 37 | 63.5 | 46.4 | **66.6** | - | - | 53.5±2.2 |
| antmaze-ld | 14.9 | 47.5 | 49.0 | 58 | 67.9 | 56.6 | 64.8 | - | - | **71.0 ±4.9** |
| Average | 50.6 | 63.0 | 70.1 | 69.3 | 79.1 | 69.6 | 78.3 | - | - | **80.6** |

provides an efficient surrogate that closely aligns with the desired consistency condition, enabling faster optimization and stronger policies.

**Variational Guidance.** We compare GTP with a baseline that combines the generative loss with a linear Q-learning actor loss. As shown in Table 3, this baseline is highly brittle: for typical coefficients ($\lambda = 0.1$ or $1.0$), training diverges due to exploding critic gradients. Even with $\lambda = 0.01$, the baseline occasionally achieves returns close to ours, but this setting is highly sensitive to the critic scale and does not transfer across tasks. In contrast, our variational guidance normalizes and clips critic signals into stable importance weights, yielding consistently high returns across seeds without per-task hyperparameter tuning. Further details and extended comparisons are provided in Appendix B.6.

Table 3: Ablation results on `hopper-medium-expert-v2` (mean ± std over 5 random seeds). Training time is wall-clock hours per run. Baselines with $\lambda = 0.1$ or $1.0$ consistently diverged.

| Method | Training Time | Score |
|---|---|---|
| **GTP (ours)** | 4.26 h | **112.2 ± 0.6** |
| w/o score approximation (ODE solver) | 5.23 h | 99.7 ± 1.7 |
| GTP-BC + linear Q-term ($\lambda = 0.01$) | 5.08 h | 111.4 ± 0.9 |
| GTP-BC + linear Q-term ($\lambda = 0.1$) | Diverged | – |
| GTP-BC + linear Q-term ($\lambda = 1.0$) | Diverged | – |

## 6 CONCLUSION

In this work, we introduced *Generative Trajectory Policies*, a new paradigm for offline RL that leverages our proposed unifying perspective of continuous-time generative ODEs. We show that while this framework offers immense expressive power, its direct application is hindered by critical challenges of computational cost, training instability, and objective misalignment. We overcame these obstacles through two theoretically principled adaptations: score approximation for efficient, stable training and a variational, advantage-weighted objective to bridge the gap between imitation and policy improvement. Our empirical results on the D4RL benchmarks validate this approach, showing that GTP establishes a new state-of-the-art for generative policies in offline RL. This work opens a promising direction for harnessing continuous-time dynamics in RL. While inference is fast, reducing the substantial training time of this model class remains an important avenue for future research.

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

CONTENTS

# A    RELATED WORK

**Expressive Policies in Offline RL**    Offline RL seeks to learn policies from static datasets, but suffers from extrapolation error when actions fall outside the dataset distribution. Early work such as Lange et al. (2012) laid the foundations for batch RL, while subsequent studies introduced more expressive policy classes, including energy-based formulations (Haarnoja et al., 2017). A major line of research has emphasized conservatism, aiming to constrain policy behavior to dataset-supported regions. Batch-Constrained Q-Learning (BCQ) (Fujimoto et al., 2019) explicitly restricted the policy to actions close to the behavior data, whereas BEAR (Wu et al., 2019) imposed a Maximum Mean Discrepancy penalty to softly align the learned policy with the dataset. Conservative Q-Learning (CQL) (Kumar et al., 2020) further advanced this idea by penalizing Q-values on out-of-distribution actions, mitigating overestimation at the cost of tuning sensitivity. In parallel, regression-based approaches sought to improve expressiveness while retaining stability. Advantage-Weighted Actor-Critic (AWAC) (Peng et al., 2019) biased behavior cloning toward high-advantage actions, while Implicit Q-Learning (IQL) (Kostrikov et al., 2022) simplified training by implicitly aligning value estimates with policy improvement, avoiding explicit constraints but limiting the ability to fully capture multimodal behaviors. Overall, these approaches highlight a persistent trade-off: policies that are more expressive tend to risk instability, while more conservative methods achieve robustness at the cost of limited representational power.

**Continuous-Time Generative Models.**    Diffusion models have significantly advanced generative modeling capabilities. Ho et al. (2020) introduced Denoising Diffusion Probabilistic Models (DDPM), utilizing forward and reverse diffusion processes to produce high-quality samples, albeit slowly. To address computational efficiency, Song et al. (2021a) developed Denoising Diffusion Implicit Models (DDIM), reducing required steps but encountering stability issues when overly accelerated. To enhance training stability, Kingma et al. (2021) integrated variational inference into diffusion models, yet this approach added complexity and computational overhead. Providing a continuous-time formulation, Song et al. (2021b) proposed modeling diffusion through Stochastic Differential Equations (SDEs), increasing flexibility but at considerable computational expense. Nichol & Dhariwal (2021) improved DDPM architectures and noise schedules, enhancing sample quality while still necessitating multiple denoising steps. Song et al. (2023) subsequently introduced Consistency Models, achieving faster sampling through trajectory consistency but potentially risking mode collapse and reduced diversity. Building on this, Kim et al. (2024) developed Consistency Trajectory Models, directly modeling diffusion trajectories via Ordinary Differential Equations (ODEs), refining sampling consistency and quality. Lastly, Frans et al. (2025) extended flow-matching models by introducing adjustable step sizes, further improving sampling efficiency without compromising quality. While this evolution has produced a powerful toolbox of trajectory-based generative models, their potential has not yet been fully realized in the RL domain.

**The Trade-off in Generative Policies**    Integrating generative models into RL allows for highly expressive policies capable of capturing complex, multimodal action distributions. Diffusion policies, first introduced by Wang et al. (2023), exemplify this potential but inherit the slow, iterative sampling of their generative counterparts. This has created a central trade-off between policy expressiveness and inference speed. One line of research aims to guide the generative process toward high-reward regions. This is achieved by incorporating energy functions (Liu et al., 2024), leveraging contrastive energy prediction (Lu et al., 2023), or weighting the diffusion objective with Q-values (Ding et al., 2024). Another major effort focuses on accelerating inference. This includes developing more efficient sampling schemes (Kang et al., 2023), distilling the diffusion policy into a fast, deterministic one (Chen et al., 2024a), and leveraging consistency models to enable few-step action generation (Ding & Jin, 2024). Recent work has focused on tightly integrating these generative policies into established RL algorithms. Examples include IDQL (Hansen-Estruch et al., 2023), which combines a diffusion policy with an IQL critic, and DPPO (Ren et al., 2025), which enables policy gradient fine-tuning. Further refinements include advanced entropy estimation techniques (Celik et al., 2025; Ma et al., 2025), trust-region guided sampling (Chen et al., 2024c), and applications to complex domains like visuomotor control (Chi et al., 2023). Despite their potential for balancing quality and speed, this emerging class of trajectory-based generative models remains largely unexplored as a policy framework, motivating our work.

# B   THEORETICAL INSIGHTS

## B.1   PRIOR MODELS AS SPECIAL CASES OF THE UNIFIED ODE FRAMEWORK

With this framework of the flow map $\Phi$, the reparameterization $\phi$, and the two fundamental training objectives, we can now clearly position and unify prior work:

**Consistency Models (CMs)**   Consistency Models (CMs) (Song et al., 2023) can be viewed as a specialized case of Eq. (2), where the flow map is restricted to $\Phi(\boldsymbol{x}_t, t, 0)$ that always targets the data origin. Their self-consistency loss corresponds directly to a Trajectory Consistency Loss: the model's prediction at time $t$ is constrained to match the prediction at a slightly earlier time $t - \Delta t$ along the same trajectory. Formally,

$$\mathcal{L}_{\text{CM}} = ||\Phi_{\boldsymbol{\theta}}(\boldsymbol{x}_t, t, 0) - \Phi_{\boldsymbol{\theta}'}(\boldsymbol{x}_{t-\Delta t}, t - \Delta t, 0)||^2, \tag{20}$$

where $\Phi_{\boldsymbol{\theta}}^-$ is the EMA model with delayed weights. Interestingly, this bootstrapping mechanism is conceptually analogous to Temporal Difference (TD) learning (Sutton, 1988), where an estimate for the present is updated to be more consistent with an estimate from the future.

**Consistency Trajectory Models (CTMs)**   CTMs (Kim et al., 2024) provide a representative example of how our unified framework manifests in prior work. Although originally motivated from a diffusion-based perspective with a PF ODE backbone, their essence lies in directly parameterizing the time-conditional flow map $\Phi(\boldsymbol{x}_t, t, s)$ in Eq. (2). The training objective can also be naturally interpreted through our lens: the self-consistency objective used in CTMs corresponds exactly to the Trajectory Consistency Loss, albeit in a specialized form where trajectories are first mapped to an intermediate time $s$ and then further mapped to the terminal time $0/\epsilon$. This design ensures that all supervision signals are anchored at the same reference time, keeping the loss consistent. In addition, the auxiliary diffusion loss serves the role of the Instantaneous Flow Loss, enforcing local fidelity along trajectories. Thus, under our framework, CTMs emerge not as an isolated construction, but as a concrete instantiation of Eq. (2) that integrates both of the fundamental training principles.

**Shortcut Models**   The work on Shortcut Models (Frans et al., 2025) also fits squarely within our unified framework in Eq.(2), focusing on learning the "average velocity" of the underlying ODE. This approach parameterizes a dedicated stepping function, $s(\boldsymbol{x}_t, t, d)$, which is trained to predict the average velocity over a future time interval of duration $d$, i.e., $[t, t + d]$. Mathematically, it learns:

$$s(\boldsymbol{x}_t, t, d) = \frac{1}{d} \int_t^{t+d} f(\boldsymbol{x}_\tau, \tau) d\tau \tag{21}$$

While Shortcut Models define their process forward in time (e.g., from $t = 0$ noise to $t = 1$ data), the principle is directly analogous to our backward-in-time formulation. Their average velocity $s$ provides a direct way to approximate our flow map $\phi(\boldsymbol{x}_t, t, t - d) = \boldsymbol{x}_t - t \cdot s(\boldsymbol{x}_t, t, d)$ (adjusting for the time direction).

Their training objectives can be understood as specific applications of our two proposed losses. First, the requirement that the learned average velocity must collapse to the instantaneous velocity in the limit $(d \to 0)$ is a direct application of our Instantaneous Flow Loss. This is implemented with a flow-matching objective that regresses the instantaneous velocity $s(\boldsymbol{x}_t, t, 0)$ against the velocity of a simple straight-line path connecting noise $\boldsymbol{x}_0$ and data $\boldsymbol{x}_1$:

$$||s_{\boldsymbol{\theta}}(\boldsymbol{x}_t, t, 0) - (\boldsymbol{x}_1 - \boldsymbol{x}_0)||_2^2 \tag{22}$$

Second, they employ a Trajectory Consistency Loss by enforcing self-consistency between steps of different durations. A step of size $2d$ is constrained to match the composition of two consecutive steps of size $d$:

$$||s_{\boldsymbol{\theta}}(\boldsymbol{x}_t, t, 2d) - s_{\text{target}}||_2^2 \tag{23}$$

where the target $s_{\text{target}}$ is constructed by bootstrapping from the model's own (stop-gradient) predictions for the two smaller steps.

However, a crucial difference in methodology emerges here. In practice, the right-hand side of Eq. (6) is treated as the supervision signal: $\Phi(\boldsymbol{x}_t, t, u)$ is obtained via an ODE solver (or its learned

approximation) and then composed forward to $s$. Our GTP framework incorporates an explicit "***teacher***" in this process to provide stable on-trajectory information. In contrast, Shortcut Models rely purely on self-consistency, where the model generates its own targets without external guidance. The viability and trade-offs of learning without such a teacher remain a critical design question, which we directly investigate through a targeted ablation study in Appendix D.4.

**Mean Flows**  Similar to Shortcut Models, the more recent Mean Flow framework (Geng et al., 2025) is also centered on learning the average velocity of the ODE. However, Mean Flows offer another perspective on our unified framework, this time through a particularly direct, first-principles formulation. At the core of this approach is the average velocity field

$$u(\boldsymbol{x}_t, r, t) = \frac{1}{t-r}\int_r^t f(\boldsymbol{x}_\tau, \tau), d\tau, \tag{24}$$

which can be seen as a reparameterization of the integral term in our flow map $\Phi$. Through this lens, the function $\phi$ in Eq. (3) can be expressed simply as $\phi(\boldsymbol{x}_t, t, s) = \boldsymbol{x}_t - t \cdot u(\boldsymbol{x}_t, t, s)$, making Mean Flows a natural instantiation of our framework.

The crucial difference lies in the training philosophy. Rather than enforcing an explicit multi-step consistency loss, Mean Flows are trained by satisfying a derived MeanFlow Identity: a local differential equation linking the average velocity $u$ to the instantaneous velocity $f$. This identity-based view is closely related to recent continuous-time consistency models (Lu & Song, 2025), which also replace explicit trajectory simulation with a local identity constraint implemented via Jacobian–vector products, thereby avoiding the need for ODE solvers. By construction, satisfying this local identity ensures that global trajectory consistency emerges as an inherent property, eliminating the need for explicit multi-step supervision.

Within our framework, this identity-based approach offers a principled alternative to consistency-based training. We therefore derive the corresponding identity for our own formulation in Appendix B.2, and empirically compare these two training philosophies—consistency versus identity—in our ablation study (Appendix D.4).

## B.2  DERIVATION OF A CONTINUOUS-TIME TRAINING OBJECTIVE

In this section, we derive an alternative training objective for our function $\phi$, inspired by the methodology of continuous-time consistency models Lu & Song (2025) and Mean Flows (Geng et al., 2025). This results in a self-contained identity that can be used for training, relying only on the function $\phi$ itself and its derivatives.

We begin with the definition of $\phi(\boldsymbol{x}_t, t, s)$ from our unified framework:

$$\phi(\boldsymbol{x}_t, t, s) = \boldsymbol{x}_t + \frac{t}{t-s}\int_t^s f(\boldsymbol{x}_\tau, \tau)\mathrm{d}\tau. \tag{25}$$

Rearranging the terms, we can isolate the integral expression:

$$\left(1 - \frac{s}{t}\right)(\phi(\boldsymbol{x}_t, t, s) - \boldsymbol{x}_t) = \int_t^s f(\boldsymbol{x}_\tau, \tau)\mathrm{d}\tau. \tag{26}$$

We differentiate both sides with respect to $t$. Then we have:

$$\frac{s}{t^2}(\phi(\boldsymbol{x}_t, t, s) - \boldsymbol{x}_t) + \left(1 - \frac{s}{t}\right)\left(\frac{\mathrm{d}\phi(\boldsymbol{x}_t, t, s)}{\mathrm{d}t} - f(\boldsymbol{x}_t, t)\right) = -f(\boldsymbol{x}_t, t). \tag{27}$$

Solving Equation 27 for $\phi(\boldsymbol{x}_t, t, s)$ yields a new identity relating the function to its own total derivative and the instantaneous vector field:

$$\phi(\boldsymbol{x}_t, t, s) = \boldsymbol{x}_t - \left(\frac{t^2}{s} - t\right)\frac{\mathrm{d}\phi(\boldsymbol{x}_t, t, s)}{\mathrm{d}t} - tf(\boldsymbol{x}_t, t). \tag{28}$$

This identity can be made fully self-referential by using the boundary condition from our Instantaneous Flow Loss, which states $\phi(\boldsymbol{x}_t, t, t) = \boldsymbol{x}_t - tf(\boldsymbol{x}_t, t)$. We use this to substitute out the $tf(\boldsymbol{x}_t, t)$ term:

$$\phi(\boldsymbol{x}_t, t, s) = \phi(\boldsymbol{x}_t, t, t) - \left(\frac{t^2}{s} - t\right)\frac{\mathrm{d}\phi(\boldsymbol{x}_t, t, s)}{\mathrm{d}t}. \tag{29}$$

For a practical implementation, the total derivative $\frac{d\phi}{dt}$ is expanded using the chain rule, noting that $\frac{d\boldsymbol{x}_t}{dt} = f(\boldsymbol{x}_t, t)$:

$$\frac{d\phi(\boldsymbol{x}_t, t, s)}{dt} = \frac{d\boldsymbol{x}_t}{dt}\partial_{\boldsymbol{x}}\phi + \partial_t\phi = f(\boldsymbol{x}_t, t)\partial_{\boldsymbol{x}}\phi + \partial_t\phi. \tag{30}$$

Substituting the vector field $f(\boldsymbol{x}_t, t) = (\boldsymbol{x}_t - \phi(\boldsymbol{x}_t, t, t))/t$ into Equation 30 and then into Equation 29 gives the final, fully-expanded form:

$$\phi(\boldsymbol{x}_t, t, s) = \phi(\boldsymbol{x}_t, t, t) - \left(\frac{t^2}{s} - t\right)\left(\frac{\boldsymbol{x}_t - \phi(\boldsymbol{x}_t, t, t)}{t}\partial_{\boldsymbol{x}}\phi + \partial_t\phi\right). \tag{31}$$

This final expression provides a self-contained regression target for $\phi(\boldsymbol{x}_t, t, s)$. A model $\phi_{\boldsymbol{\theta}}$ can be trained to satisfy this identity by minimizing the L2 distance between the left-hand and right-hand sides. The terms on the right-hand side involving $\phi$ are evaluated using the model $\phi_{\boldsymbol{\theta}}$ itself (typically with a stop-gradient), and the required partial derivatives can be computed efficiently via automatic differentiation, such as with Jacobian-vector products (JVPs).

### B.3 PROOF OF THEOREM 1

We provide a detailed proof of Theorem 1. Recall that we fix a finite time horizon $T > 0$, let $\boldsymbol{x} \sim p_{\text{data}}$, $\boldsymbol{z} \sim \mathcal{N}(0, I)$, and define $\boldsymbol{x}_t = \boldsymbol{x} + t\boldsymbol{z}$. The vector fields $f^\star, \tilde{f} : \mathbb{R}^d \times (0, T] \to \mathbb{R}^d$ are given by

$$f^\star(\boldsymbol{x}_t, t) = \frac{\boldsymbol{x}_t - \mathbb{E}[\boldsymbol{x} \mid \boldsymbol{x}_t]}{t}, \quad \tilde{f}(\boldsymbol{x}_t, t) = \frac{\boldsymbol{x}_t - \boldsymbol{x}}{t}. \tag{32}$$

The ideal and surrogate solver trajectories are denoted by

$$X_{k+1}^\star = S_{\Delta_k}[f^\star](X_k^\star), \quad \widetilde{X}_{k+1} = S_{\Delta_k}[\tilde{f}](\widetilde{X}_k), \quad X_0^\star = \widetilde{X}_0 = \boldsymbol{x}_t. \tag{33}$$

Here $S_{\Delta_k}[f] : \mathbb{R}^d \to \mathbb{R}^d$ is a one-step method of order $p$ with step size $\Delta_k = \tau_{k+1} - \tau_k$. The multi-step propagation from $t$ to $u$ is written as

$$\Psi_{t \to u}^{\text{sol}}[f] := S_{\Delta_{K-1}}[f] \circ \cdots \circ S_{\Delta_0}[f], \tag{34}$$

with maximal step $h = \max_k |\Delta_k|$. We assume throughout that $f^\star(\cdot, t)$ is Lipschitz in $x$, the solver is zero-stable, the decoder maps $\Phi_{\boldsymbol{\theta}}(\cdot, t, s)$ and $\Phi_{\boldsymbol{\theta}-}(\cdot, u, s)$ are Lipschitz in $x$, and solver states admit bounded second moments independent of $h$.

**Lemma 1** (Conditional unbiasedness). *For all $t \in (0, T]$ and $x \in \mathbb{R}^d$ with $\boldsymbol{x}_t = x$,*

$$\mathbb{E}\left[\tilde{f}(\boldsymbol{x}_t, t) \,\Big|\, \boldsymbol{x}_t\right] = \mathbb{E}\left[\frac{\boldsymbol{x}_t - \boldsymbol{x}}{t} \,\Big|\, \boldsymbol{x}_t\right] = \frac{\boldsymbol{x}_t - \mathbb{E}[\boldsymbol{x} \mid \boldsymbol{x}_t]}{t} = f^\star(\boldsymbol{x}_t, t). \tag{35}$$

*Proof.* This follows immediately from the definitions of $\tilde{f}$ and $f^\star$. $\qquad\square$

**Lemma 2** (One-step local bias). *Let $x \in \mathbb{R}^d$ be the state at time $\tau_k$. If the solver has order $p$, then*

$$S_{\Delta_k}[\tilde{f}](x) - S_{\Delta_k}[f^\star](x) = \Delta_k\big(\tilde{f}(x, \tau_k) - f^\star(x, \tau_k)\big) + O(|\Delta_k|^{p+1}). \tag{36}$$

*Consequently, conditioning on $\boldsymbol{x}_{\tau_k} = x$ and using Lemma 1,*

$$\mathbb{E}\left[S_{\Delta_k}[\tilde{f}](x) - S_{\Delta_k}[f^\star](x) \,\Big|\, \boldsymbol{x}_{\tau_k} = x\right] = O(|\Delta_k|^{p+1}). \tag{37}$$

*Proof.* By definition of an order-$p$ one-step solver, for any drift $f$,

$$S_{\Delta_k}[f](x) = x + \Delta_k f(x, \tau_k) + O(|\Delta_k|^{p+1}). \tag{38}$$

Subtracting the two cases $f = \tilde{f}$ and $f = f^\star$ gives

$$S_{\Delta_k}[\tilde{f}](x) - S_{\Delta_k}[f^\star](x) = \Delta_k\big(\tilde{f}(x, \tau_k) - f^\star(x, \tau_k)\big) + O(|\Delta_k|^{p+1}). \tag{39}$$

Taking conditional expectation w.r.t. $\boldsymbol{x}_{\tau_k} = x$ and using Lemma 1 shows that the linear term vanishes, leaving

$$\mathbb{E}\left[S_{\Delta_k}[\tilde{f}](x) - S_{\Delta_k}[f^\star](x) \,\Big|\, \boldsymbol{x}_{\tau_k} = x\right] = O(|\Delta_k|^{p+1}). \tag{40}$$

$\square$

**Proposition 1** (Global state error). *Let the solver trajectories be*

$$X^\star_{k+1} = S_{\Delta_k}[f^\star](X^\star_k), \qquad \widetilde{X}_{k+1} = S_{\Delta_k}[\tilde{f}](\widetilde{X}_k), \qquad X^\star_0 = \widetilde{X}_0 = \boldsymbol{x}_t. \tag{41}$$

*If $f^\star(\cdot, \tau)$ is Lipschitz in $x$ and the solver is zero-stable, then there exist constants $C, C'$ (independent of $h$) such that*

$$\mathbb{E}\|X^\star_{k+1} - \widetilde{X}_{k+1}\| \leq (1 + C|\Delta_k|)\, \mathbb{E}\|X^\star_k - \widetilde{X}_k\| + C'|\Delta_k|^{p+1}. \tag{42}$$

*Consequently, over the finite horizon $[0, T]$,*

$$\mathbb{E}\|X^\star_K - \widetilde{X}_K\| \leq C_T\, h^p, \tag{43}$$

*where $C_T$ depends on $T$ and the Lipschitz/stability constants but not on $h$.*

*Proof.* We start from the standard decomposition

$$X^\star_{k+1} - \widetilde{X}_{k+1} = \big(S_{\Delta_k}[f^\star](X^\star_k) - S_{\Delta_k}[f^\star](\widetilde{X}_k)\big) + \big(S_{\Delta_k}[f^\star](\widetilde{X}_k) - S_{\Delta_k}[\tilde{f}](\widetilde{X}_k)\big). \tag{44}$$

By Lipschitz regularity of $f^\star$ and zero-stability of the scheme, there exists $C > 0$ such that

$$\|S_{\Delta_k}[f^\star](x) - S_{\Delta_k}[f^\star](y)\| \leq (1 + C|\Delta_k|)\, \|x - y\|, \qquad \forall\, x, y \in \mathbb{R}^d. \tag{45}$$

For instance, for explicit Euler $S_\Delta[f](x) = x + \Delta f(x, \tau_k)$ and thus

$$\|S_\Delta[f^\star](x) - S_\Delta[f^\star](y)\| = \|(x - y) + \Delta(f^\star(x, \tau_k) - f^\star(y, \tau_k))\| \leq (1 + L_f|\Delta|)\|x - y\|. \tag{46}$$

General one-step methods admit the same bound with a (method-dependent) constant $C$.

By Lemma 2 and the law of total expectation,

$$\mathbb{E}\big\|S_{\Delta_k}[f^\star](\widetilde{X}_k) - S_{\Delta_k}[\tilde{f}](\widetilde{X}_k)\big\| = O(|\Delta_k|^{p+1}). \tag{47}$$

Combining the two parts and taking expectations yields the recursion

$$\mathbb{E}\|X^\star_{k+1} - \widetilde{X}_{k+1}\| \leq (1 + C|\Delta_k|)\, \mathbb{E}\|X^\star_k - \widetilde{X}_k\| + C'|\Delta_k|^{p+1}. \tag{48}$$

Denote $E_k := \mathbb{E}\|X^\star_k - \widetilde{X}_k\|$, $a_k := C|\Delta_k|$, $b_k := C'|\Delta_k|^{p+1}$. Then

$$E_{k+1} \leq (1 + a_k)E_k + b_k. \tag{49}$$

Unrolling the recursion gives

$$E_K \leq \Big(\prod_{j=0}^{K-1}(1 + a_j)\Big)E_0 + \sum_{i=0}^{K-1}\Big(\prod_{j=i+1}^{K-1}(1 + a_j)\Big)b_i. \tag{50}$$

Since $E_0 = 0$, the first term vanishes. For the product term, use $1 + x \leq e^x$ to obtain

$$\prod_{j=i+1}^{K-1}(1 + a_j) \leq \exp\Big(\sum_{j=i+1}^{K-1} a_j\Big) = \exp\Big(C\sum_{j=i+1}^{K-1}|\Delta_j|\Big) \leq \exp(CT) =: C_T. \tag{51}$$

For the sum, note $|\Delta_i|^{p+1} \leq h^p|\Delta_i|$, hence

$$\sum_{i=0}^{K-1} b_i \leq C' \sum_{i=0}^{K-1}|\Delta_i|^{p+1} \leq C'h^p \sum_{i=0}^{K-1}|\Delta_i| = C'h^p(t - u) \leq C'Th^p. \tag{52}$$

Combining the two bounds,

$$E_K \leq C_T \sum_{i=0}^{K-1} b_i \leq (C_T C' T)\, h^p =: \tilde{C}_T\, h^p, \tag{53}$$

which proves the claim. $\square$

**Proposition 2** (Decoder discrepancy). *If $\Phi_{\boldsymbol{\theta}-}(\cdot, u, s)$ is Lipschitz in $x$ with constant $L$, then*

$$\mathbb{E}\|\Phi_{\boldsymbol{\theta}-}(X_K^\star, u, s) - \Phi_{\boldsymbol{\theta}-}(\widetilde{X}_K, u, s)\| \leq L\,\mathbb{E}\|X_K^\star - \widetilde{X}_K\| = O(h^p), \tag{54}$$

*and, by Jensen and bounded moments,*

$$\mathbb{E}\|\Phi_{\boldsymbol{\theta}-}(X_K^\star, u, s) - \Phi_{\boldsymbol{\theta}-}(\widetilde{X}_K, u, s)\|^2 = O(h^{2p}). \tag{55}$$

*Proof sketch.* Immediate from the Lipschitz property of $\Phi_{\boldsymbol{\theta}-}$, Proposition 1, and Jensen's inequality.
$\square$

**Proposition 3** (Objective gap). *Define*

$$\mathcal{L}_{\text{ideal}}(\boldsymbol{\theta}) := \mathbb{E}\big\|\Phi_{\boldsymbol{\theta}}(\boldsymbol{x}_t, t, s) - \Phi_{\boldsymbol{\theta}-}(X_K^\star, u, s)\big\|^2, \tag{56}$$

$$\mathcal{L}_{\text{prac}}(\boldsymbol{\theta}) := \mathbb{E}\big\|\Phi_{\boldsymbol{\theta}}(\boldsymbol{x}_t, t, s) - \Phi_{\boldsymbol{\theta}-}(\widetilde{X}_K, u, s)\big\|^2. \tag{57}$$

*Then*

$$\big|\mathcal{L}_{\text{prac}}(\boldsymbol{\theta}) - \mathcal{L}_{\text{ideal}}(\boldsymbol{\theta})\big| = O(h^p). \tag{58}$$

*Proof.* Subtracting the two objectives yields

$$\mathcal{L}_{\text{prac}}(\boldsymbol{\theta}) - \mathcal{L}_{\text{ideal}}(\boldsymbol{\theta}) =$$
$$\mathbb{E}\Big[\|\Phi_{\boldsymbol{\theta}}(\boldsymbol{x}_t, t, s) - \Phi_{\boldsymbol{\theta}-}(\widetilde{X}_K, u, s)\|^2 - \|\Phi_{\boldsymbol{\theta}}(\boldsymbol{x}_t, t, s) - \Phi_{\boldsymbol{\theta}-}(X_K^\star, u, s)\|^2\Big]. \tag{59}$$

Expanding the difference gives

$$\|\Phi_{\boldsymbol{\theta}}(\boldsymbol{x}_t, t, s) - \Phi_{\boldsymbol{\theta}-}(\widetilde{X}_K, u, s)\|^2 - \|\Phi_{\boldsymbol{\theta}}(\boldsymbol{x}_t, t, s) - \Phi_{\boldsymbol{\theta}-}(X_K^\star, u, s)\|^2 =$$
$$\big\langle \Phi_{\boldsymbol{\theta}-}(X_K^\star, u, s) - \Phi_{\boldsymbol{\theta}-}(\widetilde{X}_K, u, s),\ \Phi_{\boldsymbol{\theta}-}(X_K^\star, u, s) + \Phi_{\boldsymbol{\theta}-}(\widetilde{X}_K, u, s) - 2\Phi_{\boldsymbol{\theta}}(\boldsymbol{x}_t, t, s)\big\rangle. \tag{60}$$

Applying the Cauchy–Schwarz inequality yields

$$\big|\mathcal{L}_{\text{prac}}(\boldsymbol{\theta}) - \mathcal{L}_{\text{ideal}}(\boldsymbol{\theta})\big| \leq$$
$$\sqrt{\mathbb{E}\|\Phi_{\boldsymbol{\theta}-}(X_K^\star, u, s) - \Phi_{\boldsymbol{\theta}-}(\widetilde{X}_K, u, s)\|^2} \cdot \sqrt{\mathbb{E}\|\Phi_{\boldsymbol{\theta}-}(X_K^\star, u, s) + \Phi_{\boldsymbol{\theta}-}(\widetilde{X}_K, u, s) - 2\Phi_{\boldsymbol{\theta}}(\boldsymbol{x}_t, t, s)\|^2}. \tag{61}$$

The second factor is $O(1)$ uniformly in $h$ by the bounded-moment assumption, while the first factor is $O(h^p)$ by Proposition 2. Hence the product is $O(h^p)$.
$\square$

Combining Propositions 1–3 proves that

$$\big|\mathcal{L}_{\text{prac}}(\boldsymbol{\theta}) - \mathcal{L}_{\text{ideal}}(\boldsymbol{\theta})\big| = O(h^p). \tag{62}$$

which is the claim of Theorem 1.
$\square$

### B.4 THEORETICAL RATIONALE FOR THE SCORE FUNCTION APPROXIMATION

Theorem 1 formally establishes that replacing the true vector field $f^\star$ with the surrogate $\tilde{f}$ changes the learning objective only by $O(h^p)$. Here we complement this result with intuition, showing why such an approximation is both natural and conceptually aligned with existing frameworks.

The core challenge when training our GTP from scratch is providing stable, on-trajectory supervision. The ideal consistency objective is to enforce that predictions from any two points on the same true ODE trajectory are identical. In principle, the model could generate these points itself by acting as its own "teacher" and numerically solving its learned ODE. However, when learning from scratch, this self-supervision process is inherently unstable. The initially random model produces highly inaccurate estimates of the underlying vector field, which are then used to generate its own supervision. This creates a vicious cycle where flawed targets lead to poor updates, causing further error propagation.

In contrast to our formulation with an explicit Inst Map, Consistency Training (Song et al., 2023) compensates for the missing teacher by introducing a simple analytical supervision path (a straight line) from which paired samples can be drawn. The process, illustrated in Figure 4a, is as follows:

1. A simple, straight-line supervision path (the white line) is constructed via $\boldsymbol{x}_t = \boldsymbol{x}_0 + t \cdot \boldsymbol{z}$ by sampling a data point $\boldsymbol{x}_0$ and noise $\boldsymbol{z}$.

2. A point $\boldsymbol{x}_t$ is sampled from this path. The ideal objective would require us to pair it with the corresponding point $\boldsymbol{x}_{t-\Delta t}$ on the same true ODE trajectory (the green curve). However, obtaining this point is intractable without a teacher model.

3. To create a practical objective, Consistency Training instead samples the second point, which we denote $\boldsymbol{x}'_{t-\Delta t}$, directly from the simple white line.

4. Each of these points now lies on its own true (but unknown) PF ODE trajectory. The green curve is the true trajectory passing through $\boldsymbol{x}_t$ while the blue curve is the true trajectory passing through $\boldsymbol{x}_{t-\Delta t}$.

5. The model $\Phi$ is then tasked with predicting the data origin from each of these starting points, yielding $\hat{\boldsymbol{x}}_0 = \Phi(\boldsymbol{x}_t, t, 0)$ and $\hat{\boldsymbol{x}}'_0 = \Phi(\boldsymbol{x}_{t-\Delta t}, t - \Delta t, 0)$. The consistency loss enforces that these two predictions must match, even though they originated from different points on different true trajectories:

$$\mathcal{L}_{\text{CT}} = \mathbb{E}[||\hat{\boldsymbol{x}}_0 - \hat{\boldsymbol{x}}'_0||_2^2] \tag{63}$$

The profound implication of this objective can be understood intuitively: the model is being asked to produce the same output ($\boldsymbol{x}_0$) from two different inputs ($\boldsymbol{x}_t$ and $\boldsymbol{x}'_{t-\Delta t}$). For the model to succeed at this task across all possible data points $\boldsymbol{x}_0$ and noise vectors $\boldsymbol{z}$, it cannot learn any specific set of curved trajectories. Its only viable strategy is to learn a vector field whose solution paths are, on average, straight lines that align with the simple $\boldsymbol{x}_t = \boldsymbol{x}_0 + t \cdot \boldsymbol{z}$ supervision structure. This effectively "tames" the learning problem, forcing the model to learn a consistent mapping from any point on a noisy line back to its origin.

This leads to the deeper theoretical justification. A given noisy point $\boldsymbol{x}_t$ is ambiguous; it could have been generated by many different pairs of $(\boldsymbol{x}_0, \boldsymbol{z})$. To minimize the consistency loss on average over the entire dataset, the model's output $\Phi(\boldsymbol{x}_t, t, 0)$ cannot learn to predict any single $\boldsymbol{x}_0$. Instead, it is implicitly forced to learn the optimal Bayes estimator: the conditional expectation $\mathbb{E}[\boldsymbol{x}_0|\boldsymbol{x}_t]$. This learned denoiser is precisely the component needed to define the true, underlying PF ODE. This process, where the model learns a single deterministic trajectory by averaging over all possible linear paths that could generate a point $\boldsymbol{x}_t$, is conceptually illustrated in Figure 3.

To make this connection concrete, Figure 4b illustrates the training objective of our GTP framework. While Consistency Training supervises the model indirectly by enforcing agreement between predictions from two noisy samples, our GTP formulation directly parameterizes the entire solution map $\Phi(\boldsymbol{x}_t, t, s)$ and enforces its self-consistency across intervals. Conceptually, this is nothing but the same principle as in Consistency Training: both objectives ensure that different ways of tracing back a noisy point must yield the same underlying origin. The key difference is one of perspective—Consistency Training views the model as a local denoiser (learning the map to $t = 0$), whereas GTP treats the model as the generator of a global trajectory solution (learning the map to any $s$). Thus, the two formulations are theoretically equivalent in spirit, with GTP offering a more direct and integral-level realization of the same consistency idea.

A subtle point concerns the role of the step size. In Consistency Training (Song et al., 2023), the theoretical justification is given in the limit $\Delta t \to 0$, ensuring locally correct supervision. In our setting, the step size parameter $h$ is defined as the maximum interval between adjacent time points, so the requirement $h \to 0$ is directly analogous to their $\Delta t \to 0$ condition. The difference is that our Theorem 1 makes this correspondence explicit by proving that, even for finite $h$, the surrogate objective deviates from the ideal one only by $O(h^p)$. Thus the infinitesimal-step reasoning of Consistency Training can be viewed as a special case of our more general analysis.

Finally, this principle is not an isolated finding but a direct instantiation of the same powerful idea that underpins the Flow Matching framework (Lipman et al., 2023). Both approaches learn the complex, non-linear vector field of the true ODE by supervising it with a target derived from the conditional expectation of simple, analytical paths. They differ only in what the neural network explicitly learns: while Flow Matching trains a network to regress against this expected velocity, our GTP framework learns to approximate the solution map $\Phi(\boldsymbol{x}_t, t, s)$ itself (the integral). This deep connection confirms that our training strategy is not an ad-hoc simplification but a principled and theoretically sound method for learning the correct generative dynamics.

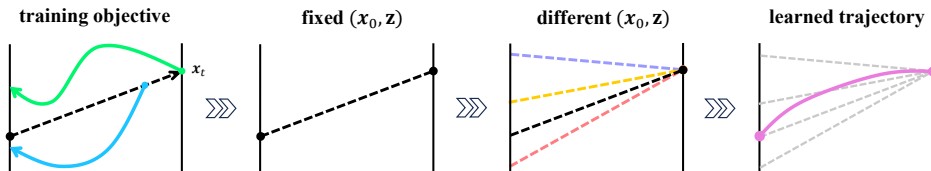

Figure 3: Illustration of the learning process. A given point $x_t$ can be formed by many different linear paths (corresponding to different pairs of $(x_0, z)$). The model is trained to learn a single, deterministic "learned trajectory" that represents the conditional expectation of these paths. This forces the model to learn the true underlying generative dynamics.

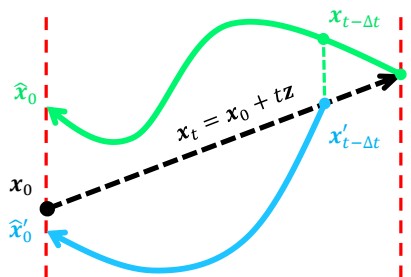

(a) Illustration of the Consistency Training objective. The process uses a simple, analytical supervision path (white line) defined by $x_t = x_0 + t \cdot z$. Two points, $x_t$ and $x'_{t-\Delta t}$, are sampled from this path. Each lies on its own true (but unknown) PF ODE trajectory (green and blue curves). The consistency loss enforces that the model's predictions for the data origin, $\hat{x}_0$ and $\hat{x}'_0$, must match, even when starting from different points on different trajectories.

(b) Illustration of the GTP Training objective. The process mirrors that of standard Consistency Training but is generalized for an arbitrary target time $s$. A student model's direct prediction from $x_t$ to a target point $x_s$ (green path) is trained to match a target model's two-step prediction via an intermediate point $x_u$ (blue path). This generalization allows the model to learn the full solution map $\Phi(x_t, t, s)$, not just the specialized map to the origin.

Figure 4: Comparison of training objectives. (a) Standard Consistency Training supervises predictions back to the origin. (b) GTP extends this principle by enforcing self-consistency across arbitrary intervals, enabling direct learning of the solution map.

### B.5 DERIVATION OF THE ADVANTAGE-WEIGHTED OBJECTIVE

This section provides the detailed theoretical derivation for the advantage-weighted learning objective presented in Section 4.2.

Our starting point is a standard objective in offline RL that seeks to maximize the Q-function while regularizing the learned policy $\pi$ to stay close to the dataset's behavior policy $\pi_{\text{BC}}$ via a KL-divergence constraint:

$$\max_\pi \quad \mathbb{E}_{a \sim \pi(\cdot|s)} \left[ Q(s, a) - \frac{1}{\eta} D_{\text{KL}}(\pi(\cdot|s) || \pi_{\text{BC}}(\cdot|s)) \right] \tag{64}$$

As shown by prior work in variational RL (Peters et al., 2010; Abdolmaleki et al., 2018; Kumar et al., 2020), the optimal solution $\pi^*$ for this problem takes the form of the behavior policy, re-weighted by the exponentiated Q-function. For greater conceptual clarity and numerical stability, this solution is typically expressed using the advantage function $A(s, a) = Q(s, a) - V(s)$:

$$\pi^*(a|s) = \frac{1}{Z(s)} \pi_{\text{BC}}(a|s) \exp(\eta A(s, a)), \tag{65}$$

where $Z(s)$ is the state-dependent normalization term. This $\pi^*$ represents the ideal, value-improved target policy we wish our model to learn. The task now becomes how to train our expressive generative

policy $\pi_{\boldsymbol{\theta}}$ to match this optimal target $\pi^*$. The natural way to do so is to minimize the KL-divergence between them:

$$\min_{\boldsymbol{\theta}} D_{\mathrm{KL}}(\pi^*(\cdot|s)||\pi_{\boldsymbol{\theta}}(\cdot|\boldsymbol{s})) = \max_{\boldsymbol{\theta}} \mathbb{E}_{a\sim\pi^*(\cdot|s)}[\log \pi_{\boldsymbol{\theta}}(\boldsymbol{a}|\boldsymbol{s})] \tag{66}$$

To compute this expectation, we use importance sampling to switch from the intractable target distribution $\pi^*$ to the tractable dataset distribution $\pi_{\mathrm{BC}}$. The importance weight is $\pi^*(a|s)/\pi_{\mathrm{BC}}(a|s) = \exp(\eta A(s,a))/Z(s)$. Substituting this into our objective gives:

$$\max_{\boldsymbol{\theta}} \quad \mathbb{E}_{(s,a)\sim\mathcal{D}}\left[\frac{\exp(\eta A(s,a))}{Z(s)} \log \pi_{\boldsymbol{\theta}}(a|s)\right] \tag{67}$$

Here we arrive at the final crucial step. The normalization term $Z(s)$ is also independent of our optimization variable $\boldsymbol{\theta}$. Therefore, when taking the gradient with respect to $\boldsymbol{\theta}$, $Z(s)$ acts as a constant scaling factor and does not affect the location of the optimum. We can thus drop it from the optimization objective, which yields the final, practical form:

$$\max_{\boldsymbol{\theta}} \quad \mathbb{E}_{(s,a)\sim\mathcal{D}}[\exp(\eta A(s,a)) \log \pi_{\boldsymbol{\theta}}(a|s)] \tag{68}$$

While we write the loss here in terms of log-likelihood for clarity, the same exponential advantage weighting directly applies to any generative training loss, including diffusion and flow-matching objectives. This confirms that applying an exponential advantage weight to the log-likelihood objective of our generative policy is the theoretically correct implementation of the variational policy optimization framework.

### B.6 Additional Discussion on Actor Loss Formulations

In Section 5.3, we compared our variational guidance against a baseline that linearly combines the generative loss with a Q-learning actor loss. Here we provide a more detailed discussion.

**Formulation.** The baseline takes the form

$$\mathcal{L}_{\mathrm{actor}} = \mathcal{L}_{\mathrm{BC}} + \lambda \mathcal{L}_Q, \tag{69}$$

while our method instead adopts a weighted behavior cloning objective:

$$\mathcal{L}_{\mathrm{weighted\text{-}BC}} = \mathbb{E}_{(s,a)\sim\mathcal{D}}\big[w(s,a)\,\mathcal{L}_{\mathrm{BC}}\big], \tag{70}$$

where $w(s,a)$ is given in Equation 14.

**Lemma 3.** *When $\mathcal{L}_{\mathrm{BC}}(\pi;\pi_\beta)$ is instantiated as a KL divergence $D_{\mathrm{KL}}(\pi_\beta(\cdot|s)\,\|\,\pi(\cdot|s))$, the linear-combination objective corresponds to a KL-regularized policy improvement whose optimizer*

$$\pi^*(a|s) = \tfrac{1}{Z(s)}\,\pi_\beta(a|s)\,\exp\big(\lambda\,Q(s,a)\big). \tag{71}$$

*Training a parametric policy $\pi_\theta$ to match $\pi^*$ then leads to a weighted-BC update*

$$\max_{\theta} \quad \mathbb{E}_{a\sim\pi_\beta(\cdot|s)}\big[\exp\big(\lambda\,Q(s,a)\big)\,\log \pi_\theta(a|s)\big], \tag{72}$$

*i.e., weighted behavior cloning with weights $w(s,a) \propto \exp(\lambda Q(s,a))$.*

*Proof Sketch.* By definition,

$$\mathcal{L}_{\mathrm{actor}} = \mathcal{L}_{\mathrm{BC}} + \lambda \mathcal{L}_Q. \tag{73}$$

When $\mathcal{L}_{\mathrm{BC}}$ is instantiated as a KL divergence and $\mathcal{L}_Q$ as the negative $Q$-expectation, this becomes

$$\mathcal{L}_{\mathrm{actor}} = \mathbb{E}_{s\sim\mathcal{D}}\Big[D_{\mathrm{KL}}\big(\pi_\beta(\cdot|s)\,\|\,\pi(\cdot|s)\big) - \lambda\,\mathbb{E}_{a\sim\pi(\cdot|s)}[Q(s,a)]\Big]. \tag{74}$$

Optimizing over $\pi$ yields

$$\pi^*(a|s) = \tfrac{1}{Z(s)}\,\pi_\beta(a|s)\,\exp\big(\lambda Q(s,a)\big), \tag{75}$$

which is exactly the same Boltzmann form derived in Appendix B.5. Minimizing the KL divergence between $\pi^*$ and the parametric policy $\pi_\theta$ is therefore equivalent to weighted BC with exponential weights. $\qquad\square$

*Remark.* Replacing $Q(s,a)$ with the advantage $A(s,a) = Q(s,a) - V(s)$ makes the weights invariant to affine shifts of $Q$; the state-value term is absorbed into $Z(s)$, leaving the training objective unchanged.

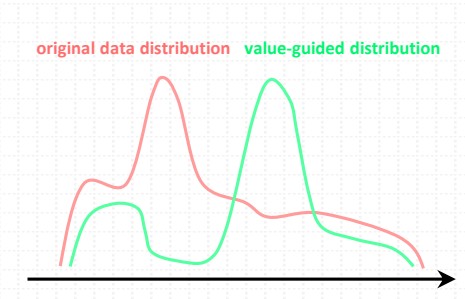

Figure 5: Illustration of weighted behavior cloning. The empirical dataset distribution (black) is reweighted into a value-guided distribution (red), which emphasizes high-value regions while remaining strictly within the data support.

**Discussion.** This result shows that, when the behavior cloning loss is instantiated as a KL divergence, the linear-combination baseline and our weighted-BC approach are consistent through the underlying KL-regularized policy improvement. For other choices of $\mathcal{L}_{\text{BC}}$, this connection is less direct. In practice, however, the raw linear form is highly sensitive to the scale of $\lambda$ and critic values, whereas our normalized and clipped weighting provides stable and robust training across settings.

Beyond this theoretical connection, the linear combination can be viewed as a more direct engineering heuristic. Its practical challenge lies in the choice of $\lambda$, which controls the relative strength of the gradient signals. Since our $\mathcal{L}_{\text{BC}}$ is already composed of multiple components, the magnitudes of $\mathcal{L}_{\text{BC}}$ and $\mathcal{L}_Q$ can differ by orders of magnitude. Stabilizing this baseline thus requires careful tuning of $\lambda$—often by matching empirical gradient norms during training—otherwise one may encounter exploding critic gradients or vanishing actor updates.

A further drawback is that the direct gradient from $\mathcal{L}_Q$ can steer the policy toward out-of-distribution (OOD) actions because it relies on critic estimates for actions with little or no dataset support. In practice, this extrapolation error propagates through the bootstrapped TD targets, inflating temporal-difference residuals and often causing the critic loss to explode—a phenomenon we frequently observed in implementation. By contrast, our weighted-BC formulation never changes the data itself: the policy is always trained on in-distribution samples, but their effective frequency (or density) is adjusted according to value estimates. This reweighting not only shifts probability mass toward high-value regions of the dataset (Figure 5), but also makes gradient magnitudes easier to control, thereby greatly reducing the occurrence of critic loss explosions in practice.

## C IMPLEMENTATION DETAILS

### C.1 EXPERIMENTAL HYPERPARAMETERS

Unless otherwise noted, all ablation studies were executed on the RTX 4090 + i9-13900K workstation; the other machines were used for main-figure experiments. All experiments are implemented in Python and conducted on five machines: one with an RTX 4090 and i9-13900K CPU (24 cores / 32 threads); three with dual A40 GPUs and dual EPYC 7313 CPUs (16 cores / 32 threads each); and one with an RTX 3050 and i7-13700H CPU (14 cores / 20 threads). RAM ranges from 24 GB to 128 GB across machines.

### C.2 DYNAMIC TIMESTEP SCHEDULING FOR ROBUST TRAJECTORY LEARNING

In our implementation, to address the trade-off between computational cost and approximation accuracy inherent in selecting the number of discretization steps $N$, we adopt a dynamic scheduling strategy inspired by Song & Dhariwal (2024). Rather than fixing $N$, the schedule gradually increases the number of steps as training progresses. This curriculum exposes the model to trajectories of varying resolutions, which prevents overfitting to a specific discretization and promotes a more faithful understanding of the underlying continuous-time dynamics. Consequently, the learned policy

Table 4: The hyperparameters in offline (including BC, $\eta = 0$) training on D4RL Gym, AntMaze, Adroit and Kitchen tasks.

| Tasks | \multicolumn{5}{c}{Hyperparameters} | | | | |
|---|---|---|---|---|---|
| | learning rate | $\eta$ | Q norm | max Q backup | gradient norm |
| halfcheetah-medium-v2 | $3 \times 10^{-4}$ | 5.0 | False | False | 9.0 |
| hopper-medium-v2 | $3 \times 10^{-4}$ | 1.0 | False | False | 9.0 |
| walker2d-medium-v2 | $3 \times 10^{-4}$ | 5.0 | False | False | 1.0 |
| halfcheetah-medium-replay-v2 | $3 \times 10^{-4}$ | 5.0 | False | False | 2.0 |
| hopper-medium-replay-v2 | $3 \times 10^{-4}$ | 5.0 | False | False | 4.0 |
| walker2d-medium-replay-v2 | $3 \times 10^{-4}$ | 5.0 | False | False | 4.0 |
| halfcheetah-medium-expert-v2 | $3 \times 10^{-4}$ | 5.0 | False | False | 7.0 |
| hopper-medium-expert-v2 | $3 \times 10^{-4}$ | 5.0 | False | False | 5.0 |
| walker2d-medium-expert-v2 | $3 \times 10^{-4}$ | 5.0 | False | False | 5.0 |
| antmaze-umaze-v0 | $3 \times 10^{-4}$ | 5.0 | False | False | 2.0 |
| antmaze-umaze-diverse-v0 | $3 \times 10^{-4}$ | 1.0 | False | True | 3.0 |
| antmaze-medium-play-v0 | $1 \times 10^{-3}$ | 5.0 | False | True | 2.0 |

becomes robust to the choice of inference steps, maintaining strong performance even with a small number of sampling steps at deployment.

Formally, the number of steps at iteration $k$ is given by:

$$N(k) = \min\left(s_0 \cdot 2^{\left\lfloor \frac{k}{K'} \right\rfloor}, s_1\right) + 1, \quad \text{where} \ \ K' = \left\lfloor \frac{K}{\log_2 \frac{s_1}{s_0} + 1} \right\rfloor. \tag{76}$$

Here $K$ is the total number of training iterations, $s_0 = 10$ and $s_1 = 1280$ are the minimum and maximum number of discretization steps, respectively. The schedule doubles the step count every $K'$ iterations until the maximum $s_1$ is reached, starting with short, computationally efficient rollouts and progressively refining toward longer, more accurate trajectories.

# D  ADDITIONAL RESULTS

## D.1  ABLATION STUDY: EFFECT OF SAMPLING HORIZON $T$

Table 5: Comparison among diffusion- and flow-based offline RL methods on D4RL Gym (mean $\pm$ std over 5 seeds).

| Gym Tasks | D-QL | QGPO | BDM | C-AC | GTP (T=5) | GTP (T=2) |
|---|---|---|---|---|---|---|
| halfcheetah-m | 51.1 | 54.1 | 57.0 | **69.1** | 53.9±0.1 | 53.1±0.5 |
| hopper-m | **90.5** | 98.0 | **98.4** | 80.7 | 90.3±2.7 | 87.8±2.3 |
| walker2d-m | 87.0 | 86.0 | 87.4 | 83.1 | 89.5±0.6 | **90.5±0.5** |
| halfcheetah-mr | 47.8 | 47.6 | 51.6 | **58.7** | 50.8±0.4 | 48.7±0.2 |
| hopper-mr | **101.3** | 96.9 | 92.7 | 99.7 | 101.7±0.3 | 101.6±0.5 |
| walker2d-mr | **95.5** | 84.4 | 89.2 | 79.5 | 94.2±0.3 | **94.3±1.4** |
| halfcheetah-me | **96.8** | 93.5 | 93.2 | 84.3 | 93.8±0.8 | **96.2±0.4** |
| hopper-me | 111.1 | 108.0 | 104.9 | 100.4 | **112.2±0.6** | 111.7±0.6 |
| walker2d-me | 110.1 | 110.7 | 111.1 | 110.4 | **114.2±0.3** | **114.2±1.0** |
| **Average** | 87.9 | 86.6 | 87.3 | 85.1 | **89.0** | 88.7 |

The ablation results in Table 5 show that using a shorter sampling horizon ($T = 2$) yields essentially the same performance in expectation as the default $T = 5$ across all Gym tasks. While the $T = 2$ scores exhibit slightly higher variance, the means remain very close to those of $T = 5$, indicating no meaningful degradation in policy quality. Importantly, reducing the horizon from $T = 5$ to $T = 2$ leads to a substantial improvement in efficiency, as it requires significantly fewer sampling steps while

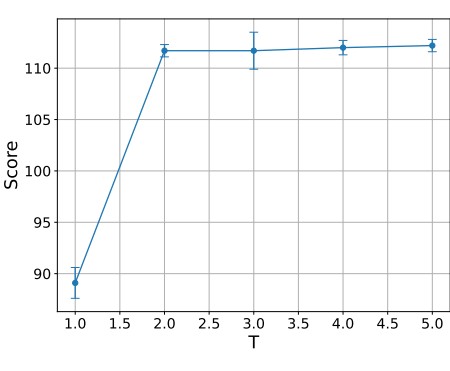 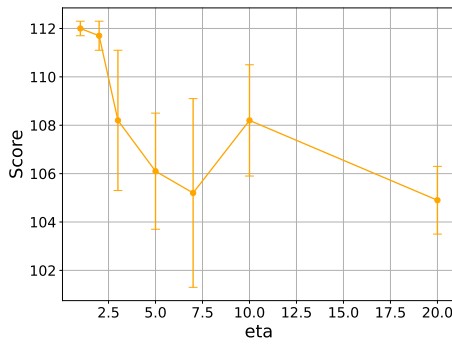

(a) Effect of sampling horizon $T$.  (b) Effect of advantage temperature $\eta$.

Figure 6: Sensitivity analysis of GTP on `hopper-medium-expert-v2`. Shaded regions denote $\pm$ one standard deviation over 5 seeds.

maintaining comparable returns. This demonstrates that GTP does not rely on long ODE trajectories for strong performance, and that a very short generative trajectory is sufficient for effective policy improvement in offline RL.

### D.2 ABLATION STUDY: SENSITIVITY ANALYSIS

To further examine the robustness of GTP, we conduct a sensitivity analysis with respect to the sampling horizon $T$ and the advantage temperature $\eta$ in the value-guided objective.

**Sampling horizon $T$.** Figure 6(a) plots the return on `hopper-medium-expert-v2` as a function of the sampling horizon $T \in \{1, 2, 3, 4, 5\}$. Performance improves substantially when increasing $T$ from 1 to 2, but quickly saturates for $T \geq 2$, with nearly indistinguishable returns for $T = 2, 3, 4, 5$. This confirms that GTP does not rely on long ODE trajectories: a very short horizon ($T = 2$) is already sufficient in practice.

**Advantage temperature $\eta$.** Figure 6(b) shows the effect of varying the advantage temperature $\eta \in \{1, 2, 3, 5, 7, 10, 20\}$. GTP is relatively stable for $\eta$ in the range $[1, 3]$, with a mild peak around $\eta = 1$–$2$. Larger values of $\eta$ slightly degrade performance due to over-emphasizing a small number of high-advantage samples, but the overall variation remains moderate. These results indicate that GTP is robust to the choice of $\eta$ over a reasonably wide range and does not require fine-tuning of this hyperparameter.

### D.3 ABLATION STUDY: EFFICIENCY–PERFORMANCE TRADE-OFF AT INFERENCE

A core claim of our work is that GTP resolves the trade-off between expressiveness and efficiency. To validate this, we compare inference-time efficiency across different generative policy classes on `halfcheetah-medium-expert-v2`. Specifically, we sample 100 trajectories and report the average wall-clock time per inference step together with final policy performance. We benchmark GTP against diffusion policies with $T = 5$ sampling steps and consistency models with $T = 2$ steps, while also evaluating GTP with $T = 5$ steps to ensure a fair comparison in terms of sampling cost.

**Results and Analysis.** Table 6 summarizes the results. We find that GTP with $T = 5$ achieves slightly faster inference than diffusion with the same number of steps, while delivering substantially better performance. Using a shorter sampling horizon also improves efficiency: GTP with $T = 2$ uses fewer sampling steps while maintaining nearly the same performance as $T = 5$. Compared to consistency models with $T = 2$, our method is moderately slower but significantly more expressive, closing the gap between efficiency and policy quality. Together, these results show that GTP provides a flexible and favorable trade-off: it can match consistency-level efficiency when using $T = 2$, while retaining the strong performance benefits of the full GTP architecture. This demonstrates that

GTP provides a favorable balance between efficiency and performance, resolving a long-standing limitation of prior generative policies.

Table 6: Ablation results on `halfcheetah-medium-expert-v2`. Inference time is averaged over 100 sampled trajectories. GTP achieves a superior trade-off, being faster than diffusion while outperforming consistency.

| Method | Inference Time (ms) |
|---|---|
| Diffusion Policy ($T = 5$) | 1.16 |
| Consistency Model ($T = 2$) | 0.55 |
| GTP ($T = 5$) | 0.94 |
| GTP ($T = 2$) | 0.67 |

### D.4 ABLATION ON ACTOR LOSS FORMULATION

We further examined whether our explicit teacher is truly necessary by testing two teacher-free philosophies: (i) the self-consistency principle underlying Shortcut Models (Frans et al., 2025), and (ii) the identity-based formulation of continuous consistency models and Mean Flows (Lu & Song, 2025; Geng et al., 2025). Both yield a continuous-time actor loss that is theoretically elegant and fully self-referential, requiring no external supervision.

**Results and Analysis.** In practice, however, both variants were unsatisfactory. The identity-based loss demanded repeated JVPs in PyTorch, leading to prohibitive memory and runtime overhead. Despite multiple attempts and stabilization tricks, all identity-based runs eventually encountered severe divergence, with either actor loss blowing up or critic loss exploding due to OOD data. We nevertheless completed several such runs: one experiment took 6.03 hours to finish but still collapsed in performance. We suspect this discrepancy arises because the original Mean Flows were implemented on TPUs with specialized auto-differentiation, while PyTorch+GPU implementations incur heavy JVP costs and suffer from numerical instability—issues that are especially problematic in RL, where stable training is crucial. The self-consistency variant, while computationally lighter, also produced unstable targets and degraded policy quality.

In contrast, our teacher-guided score approximation provided stable, efficient training and consistently stronger policies. These results highlight that while teacher-free objectives are conceptually appealing, they are not yet practical under standard GPU-based RL settings. We leave further improvements in this direction as an interesting avenue for future work.

### D.5 VISUALIZING EXPRESSIVENESS AND EFFICIENCY IN MULTI-GOAL ENVIRONMENTS

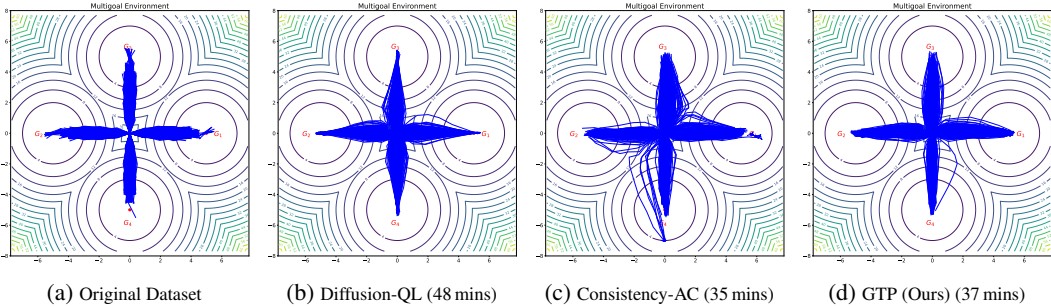

| (a) Original Dataset | (b) Diffusion-QL (48 mins) | (c) Consistency-AC (35 mins) | (d) GTP (Ours) (37 mins) |

Figure 7: Policy visualization in a 2D multi-goal environment.

To provide an intuitive, visual confirmation of our claims, we design a 2D multi-goal environment where the optimal policy is inherently multi-modal. As shown in Figure 7, our GTP model accurately captures the four distinct modes of the data, learning a policy that successfully reaches all goals. In contrast, while Diffusion-QL also captures the modes, it does so at a higher computational cost. Consistency-AC is faster but fails to capture all modes, suffering from degraded policy quality. This

Table 7: Ablation results on `hopper-medium-expert-v2` (mean ± std over 5 random seeds). Teacher-free objectives are either too costly (Mean Flows) or unstable (Shortcut), while our teacher-based formulation achieves the best trade-off.

| Method | Training Time | Score |
|---|---|---|
| Shortcut Models (no teacher) | 4.58 h | $76.1 \pm 5.7$ |
| Mean Flows (identity) | 6.03 h | Diverged |
| **GTP (ours)** | 4.26 h | $\mathbf{112.2 \pm 0.6}$ |

visualization provides a clear illustration of our method's central achievement: successfully modeling diverse, multi-modal behaviors without sacrificing computational efficiency.

