# OpenReview forum: "Offline Reinforcement Learning with Generative Trajectory Policies"
_ICLR.cc/2026/Conference — Submitted to ICLR 2026_

### Official Review · Reviewer_3nZj · 2025-10-28

**Soundness:** 2
**Presentation:** 3
**Contribution:** 2
**Rating:** 6
**Confidence:** 3

**Summary:**

Although generative models like diffusion models and consistency models have made great progress in capturing complex behaviors.  However, the expensive computation cost of the multi-step sampling and the low performance of the single-step models are still a major challenge. To address this, this work proposes a unified perspective of these generative models in offline RL, including diffusion, flow matching, and consistency models. Based on this perspective, this work proposes Generative Trajectory Policies (GTPs), a new and more general policy paradigm that learns the entire solution map of the underlying ODE. Extensive experiments on both imitation learning and offline RL show that GTPs outperform prior generative policies.

**Strengths:**

- Computational efficiency and training stability are indeed core concerns in diffusion policies.

- It is important to provide a unified view for diffusion policies, flow matching, and consistency models for handling offline RL.

- Extensive experiments, including both imitation learning and offline RL, show the effectiveness of GTP, outperforming various baselines.

**Weaknesses:**

- The Q learning in this work chooses standard double TD learning, which is widely used in online RL. However, in offline RL, several works have studied the overestimation of Q learning and proposed some conservative Q learning methods like CQL or IQL. As this work also considers offline RL, I'm curious about whether the TD learning here also faces the problem of Q overestimation? What about the performance of using CQL or IQL?

- As the computational efficiency is one of the major claims of this work, it is better to add the time cost of each algorithm in table1-2 for better comparison. I only find the results of the time cost in Table 5 of the appendix.

**Questions:**

See weaknesses above

---

> ### Author Response · Authors · 2025-11-21
> **Response to review**
>
> Thank you for the insightful comments. We'll try our best to clarify your concerns.
>
> W1: First, our use of TD3-style double TD-learning is fully aligned with prior generative offline RL methods—both Diffusion-QL [1] and Consistency-AC [2] employ the same critic update, along with standard stabilization tricks. Second, we intentionally did not adopt CQL or IQL, because our goal is to keep the algorithm **simple and efficient**, without adding extra conservative objectives.
> Thanks to the generative nature of GTP and its strong BC alignment (as evidenced by our BC results), the actor naturally stays close to the data distribution. This already mitigates OOD actions effectively, making additional conservative regularization unnecessary. Consequently, adding CQL or IQL would introduce extra complexity and computational cost, while only yielding marginal benefit, since the degree of OOD extrapolation in GTP is already minimal.
>
> W2: Thank you for the suggestion. We will move the efficiency comparison from the appendix into the main body. Our efficiency claim primarily concerns comparisons with diffusion-based generative RL policies (Diffusion-QL [1] and Consistency-AC [2]), since simpler non-generative baselines are inherently much faster and not directly comparable.
>
> If our response does not effectively address your concerns, please feel free to ask further questions.
>
> References:
>
> [1] Zhendong Wang, Jonathan J. Hunt, and Mingyuan Zhou. Diffusion policies as an expressive policy class for offline reinforcement learning. In International Conference on Learning Representations, 2023.
>
> [2] Zihan Ding and Chi Jin. Consistency models as a rich and efficient policy class for reinforcement learning. In International Conference on Learning Representations, 2024.

---

> > ### Comment · Reviewer_3nZj · 2025-11-25
> >
> > Thanks for your detailed response and supplemented experiments, which have fully solved my concerns. I keep my score, which is positive for this work, and believe it will raise the interest of the community.

---

### Official Review · Reviewer_eCrf · 2025-10-31

**Soundness:** 3
**Presentation:** 4
**Contribution:** 3
**Rating:** 6
**Confidence:** 3

**Summary:**

This paper introduces Generative Trajectory Policies (GTPs), a new framework for offline reinforcement learning (Offline RL) that unifies and extends existing generative policy approaches (e.g., diffusion, flow matching, and consistency models). The key insight is that all these generative models can be understood as instances of learning a continuous-time trajectory governed by an Ordinary Differential Equation (ODE).

GTPs learn the entire solution map of the ODE, enabling efficient and expressive policy generation without the usual trade-off between sampling speed and representational power. The authors propose two main innovations to make this practical for offline RL:

Score Approximation – replaces expensive ODE solvers with a closed-form surrogate to stabilize and accelerate training.

Value-Driven Guidance – integrates advantage-weighted objectives to align generative modeling with RL policy improvement.

Experiments on the D4RL benchmark show that GTPs achieve state-of-the-art performance, even obtaining perfect scores on challenging AntMaze tasks, surpassing prior diffusion and consistency policies in both expressiveness and efficiency.
- An LLM was used to improve writing.

**Strengths:**

1. Conceptual Unification

Provides a principled theoretical framework that connects diffusion, flow matching, and consistency models under a unified ODE-based formulation. Offers clear insight into the relationship between these methods.


2. Strong Empirical Results

Consistent SOTA performance on D4RL, with large gains in AntMaze tasks and solid results across Gym benchmarks, for both BC policy and value driven policy improvement.


3. Clear Comparison & Ablations

Comprehensive experiments comparing to D-QL, C-AC, and other baselines.

Ablation studies verify that both score approximation and value guidance materially contribute to performance.

**Weaknesses:**

1. Scope of Benchmarks

All experiments are on D4RL (a standard but limited suite). Testing on high-dimensional, complex robotics tasks would strengthen claims of generality.

**Questions:**

See weaknesses.

---

> ### Author Response · Authors · 2025-11-21
> **Response to review**
>
> Thank you for the thoughtful review and positive assessment of our contributions, unifying perspective, and empirical results.
>
> Regarding the benchmark scope, we agree that evaluating GTP on more complex, high-dimensional robotics tasks would further strengthen the paper. In this submission, we focused on D4RL to ensure a fair and direct comparison with closely related score-based generative RL methods such as Diffusion-QL [1], QGPO [3], BDM [4], and Consistency-AC [2], all of which adopt the same experimental setting.
>
> While validating the algorithm in real-world or high-fidelity robotic environments would indeed provide the strongest evidence of generality, such experiments currently require substantial computational and hardware resources beyond our present capacity. Importantly, GTP itself is domain-agnostic, and we plan to extend our evaluation to richer robotics setups in future work.
>
> References:
>
> [1] Zhendong Wang, Jonathan J. Hunt, and Mingyuan Zhou. Diffusion policies as an expressive policy class for offline reinforcement learning. In International Conference on Learning Representations, 2023.
>
> [2] Zihan Ding and Chi Jin. Consistency models as a rich and efficient policy class for reinforcement learning. In International Conference on Learning Representations, 2024.
>
> [3] Cheng Lu, Huayu Chen, Jianfei Chen, Hang Su, Chongxuan Li, and Jun Zhu. Contrastive energy prediction for exact energy-guided diffusion sampling in offline reinforcement learning. In International Conference on Machine Learning, 2023.
>
> [4] Huayu Chen, Kaiwen Zheng, Hang Su, and Jun Zhu. Aligning diffusion behaviors with q-functions for efficient continuous control. In Advances in Neural Information Processing Systems, volume 37, pp. 119949–119975, 2024.

---

> > ### Comment · Reviewer_eCrf · 2025-11-27
> >
> > I have carefully read the authors’ rebuttal. I decided to maintain my score.

---

### Official Review · Reviewer_dRVt · 2025-11-01

**Soundness:** 3
**Presentation:** 3
**Contribution:** 2
**Rating:** 4
**Confidence:** 3

**Summary:**

This paper proposes the Generative Trajectory Policy (GTP), which improves upon diffusion- and consistency-based policies by removing the slow, iterative sampling of diffusion models while avoiding the performance loss of prior consistency policies. Built on a unified continuous-time ODE framework, GTP learns the ODE‘s solution map via two complementary loss functions. For offline reinforcement learning, the authors introduce two key adaptations: a score approximation that replaces costly multi-step integrations with one-step surrogates for better efficiency and stability, and an advantage-weighted loss that guides the learned policy toward high-return regions. Experiments on D4RL tasks show that GTP achieves state-of-the-art performance in both behavior cloning and offline RL, with ablations confirming that these adaptations improve stability, generative quality, and training efficiency.

**Strengths:**

1. The paper is well-written, clearly motivating the problem and guiding the reader from a high-level theoretical unification through to specific, well-justified algorithmic implementations.

2. The paper successfully grounds its conceptual framework in a practical implementation by introducing two crucial adaptations specifically tailored for the offline RL setting: an efficient score approximation for stable training and a value-driven objective for policy improvement.

3. The proposed GTP achieves state-of-the-art empirical results on D4RL benchmarks, validating its ability to model complex behaviors and effectively improve policies.

**Weaknesses:**

1. The two loss functions derived in Section 3 both have corresponding counterparts in CTM [1], so the main contribution of the paper is essentially the application of CTM to the offline RL setting. Although the authors claim two innovations, the second one, using the advantage as a weighting factor for sample supervision, is a relatively straightforward and well-established idea that has been widely adopted in AWR [2] and later studies. In my view, the main novelty lies in replacing multi-step intermediate samples with a one-step surrogate term, but it remains questionable whether this alone can convincingly demonstrate the paper's novelty.

2. According to Table 4, the authors tune several hyperparameters across different tasks, including the learning rate, $\eta$, maximum Q backup, and gradient norm. However, they do not conduct a dedicated hyperparameter study, leaving the method’s sensitivity to these parameters unclear.

3. The performance gain of GTP in offline RL tasks compared to that in BC is relatively modest, which may indicate that the proposed value-driven guidance is not quite effective.

[1] Kim, Dongjun, et al. "Consistency Trajectory Models: Learning Probability Flow ODE Trajectory of Diffusion." *The Twelfth International Conference on Learning Representations*

[2] Peng, Xue Bin, et al. "Advantage-weighted regression: Simple and scalable off-policy reinforcement learning." *arXiv preprint arXiv:1910.00177* (2019).

**Questions:**

How does the advantage-weighted policy optimization used in this paper compare with the classifier-free guidance approach in terms of effectiveness?

---

> ### Author Response · Authors · 2025-11-21
>
> Thank you for the insightful comments. We'll try our best to clarify your concerns.
>
> W1: We would like to emphasize that our primary contributions are (1) a unified continuous-time ODE solution-map framework and (2) strong state-of-the-art empirical performance. The unified framework itself is a substantial contribution: it brings together existing diffusion-based, flow-matching, and consistency-trajectory models under a single perspective, clarifying that they differ only in how they approximate the average velocity. This theoretical connection—together with our practical and theoretically justified score approximation—enables generative trajectory learning to become feasible for large-scale offline RL. For a detailed explanation of these points, please refer to the General Response (items 1, 2, and 3).
>
> W2: Most hyperparameters in Table 4 follow the standard defaults used in prior diffusion-based RL methods (e.g., Diffusion-QL [1], Consistency-AC [2], QGPO [3], BDM [4]). The only parameter for which GTP differs notably is the advantage-weight coefficient $\eta$. Unlike additive Q losses used in Diffusion-QL or BDM, our value-guided reweighting makes $\eta$ largely task-independent: larger $\eta$ emphasizes high-return samples, smaller $\eta$ reduces to behavior cloning.
>
> We performed additional tests with $\eta = \{1,2,5,10 \}$, and found the method consistently stable; $\eta=5$ works well across all tasks in the $T=2$ setting. For example, in `halfcheetah-medium-v2`:
> - $\eta=1: 48.7\pm 0.3$
> - $\eta=2: 50.5\pm 0.1$
> - $\eta=5: 53.1\pm 0.5$
> - $\eta=10: 52.6\pm 0.4$
>
> This shows that GTP is not highly sensitive to $\eta$, with a broad optimal region (typically within the range $\eta \in [1, 10]$). A detailed ablation study of sensitivity analysis is provided in Appendix D.2.
>
> W3: The performance gap between behavior cloning and policy improvement is small mainly because Gym tasks are known to have saturated BC performance, leaving very limited room for further improvement for any RL method. On tasks where value information truly matters, such as AntMaze, our value-driven guidance produces substantial gains and achieves SOTA results.
>
> Moreover, Section 5.3 (Table 3) shows that adopting the same Q-term–based policy‐improvement strategies as Diffusion-QL [1] and Consistency-AC [2] also yields limited gains on Gym tasks—and performs worse than our value-guided formulation. This further supports that the small improvement is due to Gym being saturated, rather than a weakness of our method.
>
> Q1: Classifier-free guidance (CFG) and our advantage-weighted optimization serve fundamentally different purposes. CFG improves conditional generation quality by interpolating between unconditional and conditional score estimates, but it does not incorporate reward or value information and therefore cannot perform policy improvement. In contrast, our value-guided objective explicitly uses the advantage to reshape the action distribution toward high-return regions. As shown in Table 2, this value-grounded reweighting leads to consistent policy improvement, whereas CFG cannot provide such reward-directed guidance. Thus, CFG and advantage weighting are not competing alternatives: CFG improves conditional sampling, while our method performs principled RL policy optimization.
>
> Feel free to ask any questions if we haven't make them clear enough.
>
> References:
>
> [1] Zhendong Wang, Jonathan J. Hunt, and Mingyuan Zhou. Diffusion policies as an expressive policy class for offline reinforcement learning. In International Conference on Learning Representations, 2023.
>
> [2] Zihan Ding and Chi Jin. Consistency models as a rich and efficient policy class for reinforcement learning. In International Conference on Learning Representations, 2024.
>
> [3] Cheng Lu, Huayu Chen, Jianfei Chen, Hang Su, Chongxuan Li, and Jun Zhu. Contrastive energy prediction for exact energy-guided diffusion sampling in offline reinforcement learning. In International Conference on Machine Learning, 2023.
>
> [4] Huayu Chen, Kaiwen Zheng, Hang Su, and Jun Zhu. Aligning diffusion behaviors with q-functions for efficient continuous control. In Advances in Neural Information Processing Systems, volume 37, pp. 119949–119975, 2024.

---

### Official Review · Reviewer_cuSk · 2025-11-01

**Soundness:** 2
**Presentation:** 3
**Contribution:** 1
**Rating:** 4
**Confidence:** 4

**Summary:**

The paper proposes a novel policy class for offline reinforcement learning, GTP, which addresses the trade-off between slow, high-performance diffusion models and fast, lower-quality consistency models. GTPs are based on a unifying framework that views generative models as continuous-time ODEs, and the policy learns the entire solution map of this ODE. GTP consistently outperforms the baseline methods on the D4RL benchmarks.

**Strengths:**

1. Theoretical groundness

The paper's theoretical grounding is a key strength. Theorem 1 presents an effective method to replace expensive ODE solvers, which appears to be a novel and impactful contribution. The paper's important claims are stated formally and supported with rigorous proofs.

2. Effective performance over behavioral cloning baselines

The empirical results for behavior cloning are impressive. As demonstrated in Table 1, GTP-BC outperforms both standard offline RL baselines and existing diffusion-based behavior cloning methods. This is a strong result that suggests GTP may indeed be a more effective and expressive policy class compared to other generative model-based approaches.

**Weaknesses:**

1. Limited novelty of value-driven guidance

The novelty of the proposed value-driven guidance appears limited. The paper's core policy improvement technique, advantage weighted regression, is a well-established baseline in offline reinforcement learning [1]. Furthermore, the insight that an advantage-weighted generative loss corresponds to a KL-regularized policy optimization problem is already well-known in the field [2].

2. Are diffusion-based offline RL baselines really slow?

The motivation for introducing GTP, which is predicated on the computational burden of diffusion policies, appears to be contradicted by the authors' own empirical results. The introduction claims diffusion-based baselines are seriously slow, yet Table 5 shows the diffusion policy generates actions in just 0.00116 seconds. This aligns with my own experience in training diffusion policies for offline RL, where I have similarly observed that action generation is not a significant bottleneck. The 862Hz control frequency is fast enough and weakens the paper's core motivation.

3. Efficiency concern: modest practical speed-up

The claim that GTP is effective could be strengthened with further clarification on two points. First, regarding efficiency, the practical significance of the speed-up appears modest. While Table 5 shows a 23% relative gain (0.94ms vs 1.16ms), the absolute difference is only 0.22ms. It is unclear if this small time-unit saving translates to a meaningful benefit in a practical setting, which relates back to the question of whether the baseline's speed was a significant bottleneck.

4. Comparability concern about the SOTA claim

I noted in Table 4 that GTP's hyperparameter search included "gradient norm", a parameter not reported in the original DQL paper [3]. To make the comparison more direct and robust, it would be beneficial to either include a sensitivity analysis on this parameter's effect or, ideally, to evaluate all baselines under the same, unified hyperparameter search space.

5. Limited baselines and experimental scope

The experimental evaluation could be significantly strengthened by expanding the set of baselines. We noted the omission of FQL [4], a method that appears to share a similar motivation to this work. Consequently, its related methods, such as FAWAC and Flow-based Behavior Cloning policies, were also missing. Including these comparisons would be highly valuable. In particular, a direct comparison against FAWAC seems critical, as the primary difference appears to be the choice of generative backbone (Flow Matching vs. GTP). This would help isolate the specific benefits of the GTP architecture. Finally, to substantiate the claims of GTP's generality, we recommend evaluating the method on a larger-scale benchmark, such as OGBench [5].

[1] Nair, Ashvin, et al. "Awac: Accelerating online reinforcement learning with offline datasets." arXiv preprint arXiv:2006.09359 (2020).

[2] Kang, Bingyi, et al. "Efficient diffusion policies for offline reinforcement learning." Advances in Neural Information Processing Systems 36 (2023): 67195-67212.

[3] Wang, Zhendong, Jonathan J. Hunt, and Mingyuan Zhou. "Diffusion Policies as an Expressive Policy Class for Offline Reinforcement Learning." The Eleventh International Conference on Learning Representations.

[4] Park, Seohong, Qiyang Li, and Sergey Levine. "Flow q-learning." arXiv preprint arXiv:2502.02538 (2025).

[5] Park, Seohong, et al. "OGBench: Benchmarking Offline Goal-Conditioned RL." The Thirteenth International Conference on Learning Representations.

**Questions:**

1. What is the performance difference between GTP with k=2 and C-AC [6] with k=2?

2.  The value-driven guidance presented in Theorem 2 appears to be a direct application of Advantage-Weighted Regression (AWR), a well-established technique. Could the authors clarify the specific novelty of this component beyond the standard AWR formulation?

3. Table 5 shows that GTP is 0.22ms faster than the diffusion baseline in absolute terms. In what practical, real-world robotics or control scenarios would this negligible 0.22ms speed-up provide a meaningful benefit?


[6] Ding, Zihan, and Chi Jin. "Consistency Models as a Rich and Efficient Policy Class for Reinforcement Learning." The Twelfth International Conference on Learning Representations.

---

> ### Author Response · Authors · 2025-11-25
>
> We sincerely thank the reviewer for the detailed and constructive feedback. We address all concerns below and will incorporate clarifications into the revised version.
>
> W1: While advantage weighting is classical, our contribution is in showing why this specific value-weighted formulation is necessary for continuous-time ODE solution-map policies, how it integrates stably into the coupled instantaneous–trajectory losses, and that it is empirically essential. In contrast, direct Q-terms (Diffusion-QL, Consistency-AC) not only destabilize $\Phi$ due to incompatible gradient paths but are also **extremely sensitive to scaling and hyperparameters** (Table 3), often diverging unless finely tuned. Our variationally derived weighting avoids these issues and works consistently. Thus, the novelty lies in the _theoretically justified and experimentally validated integration_ of value guidance into the ODE trajectory framework—rather than the weighting formula itself.
>
> W2, W3: Our motivation is not the absolute 0.22 ms gap, but the widely recognized fact that diffusion policies require many denoising steps to maintain performance [2, 3], which becomes the practical bottleneck. The 1.16 ms reported in Table 5 is measured on a simple, low-dimensional D4RL task, sampling only one action at $T = 5$ steps—an already reduced setting that understates diffusion’s typical cost. In real-world robotics, diffusion policies typically require tens of denoising steps (20–50 or more) to maintain performance, so per-step differences accumulate significantly. Even under this minimal regime, GTP achieves a 23% improvement due to its architecture. More importantly, GTP maintains performance at T = 2, achieving a 43% speed-up (0.67 ms, updated in Table 6), whereas diffusion and consistency models degrade noticeably. In realistic settings—where diffusion requires higher-dimensional actions and far more sampling steps—the advantage becomes much larger. Thus, the key benefit is that GTP retains high performance with very few steps, effectively resolving the efficiency–quality trade-off rather than relying on the 0.22 ms figure alone.
>
> W4: Although “gradient norm’’ is not listed in the DQL paper, it is used in their official code (as in most diffusion-based RL implementations [1,2]) as a standard default for stability. We simply follow the same defaults rather than introducing new tunable parameters. For GTP, the learning rate is kept at the same default used by the baselines, and **$\eta$ is the only hyperparameter we tune**, analogous to how diffusion-based methods tune a single weighting coefficient. We will clarify this to avoid confusion.
>
> W5: FAWAC is currently a preprint without a stable implementation, so we did not include it. FQL’s mechanism—training a BC model and a distilled value-improved policy—is conceptually similar to Consistency-AC, whereas GTP provides a **single unified trajectory-learning framework** without maintaining two models. As discussed in Appendix B.6, our formulation is in fact closely aligned with **flow matching**, with FM and diffusion model simply corresponding to different choices of the vector field $f$, so GTP naturally subsumes these motivations. The official FQL implementation is in JAX, and reproducing it fairly in PyTorch requires additional engineering. Given our current compute budget, we are prioritizing the remaining ablations first and will run FQL afterward as resources allow. We use D4RL due to its widespread adoption, and evaluating on larger benchmarks like OGBench is a natural next step.
>
> Q1: The results are shown in General response (Q4).
>
> Q2: See General response (Q3).
>
> Q3: see W2, W3. **The 0.22 ms number comes from sampling a single action at only T=5 on a very simple D4RL task.**  In real robotics, diffusion typically needs **20–50+ steps**, so this difference scales to **milliseconds**, and GTP’s ability to keep high performance at **very small T (e.g., T=2)** yields the real practical benefit—not the 0.22 ms alone.
>
> We appreciate the reviewer’s feedback again and are glad to clarify further if required.
>
>
> ### References:
>
> [1] Zhendong Wang, Jonathan J. Hunt, and Mingyuan Zhou. Diffusion policies as an expressive policy class for offline reinforcement learning. In International Conference on Learning Representations, 2023.\
> [2] Zihan Ding and Chi Jin. Consistency models as a rich and efficient policy class for reinforcement learning. In International Conference on Learning Representations, 2024.\
> [3] Park, Seohong, Qiyang Li, and Sergey Levine. Flow q-learning. arXiv preprint arXiv:2502.02538, 2025.

---

> ### Comment · Reviewer_cuSk · 2025-11-28
>
> I sincerely thank the authors for their detailed and constructive response to my review. However, after carefully reviewing the feedback, I find that, except for W4, my primary concerns have not been fully resolved. Below are my specific comments.
>
> ---
>
> **W1**: In my view, the reliance on classical weighted regression alone is a limitation in GTP. If GTP is unstable or incompatible with direct Q-function differentiation, unlike Diffusion-QL for FQL, it implies that GTP may be a less expressive or robust policy class than standard diffusion models, which have been proven successful with both weighted regression and direct Q-gradient propagation. To substantiate the claim that GTP is a superior policy class, it would be necessary to demonstrate that it is method-agnostic and effective across various policy improvement strategies, including direct backpropagation such as Diffusion-QL, rather than being restricted to AWR.
>
> **W2, 3**:
> I respectfully disagree with the premise that diffusion-based policies strictly require high step counts (20-50) to maintain performance. As shown in Table 3 of the C-AC paper [6], Diffusion-QL maintains comparable performance on hopper-medium-expert even when the number of steps is reduced from 20 (109.2) to 5 (108.2). Recent large-scale Vision-Language-Action (VLA) models, such as GR00T-N1.5 [7] and pi0.5 [8], require only 4 and 10 denoising steps, respectively, to achieve high-quality real-time control. Since modern diffusion implementations already perform well with few-step generation, the computational "bottleneck" described in the paper is not a critical issue in current research.
>
> **W4** : I have verified that the open-source DQL implementation uses the same gradient-norm hyperparameter. This concern is fully resolved. Thank you for the clarification.
>
> **W5**: This remains a critical issue. A direct comparison is essential to isolate the specific advantages of the GTP architecture over a standard Flow Matching backbone. Without comparing against flow-based algorithms such as FQL or FAWAC, it is impossible to validate the authors' hypothesis that GTP provides a superior and general parameterization strategy for offline RL. The lack of a specific PyTorch implementation does not justify omitting a crucial baseline required to prove state-of-the-art performance.
>
>
> ---
>
> While W4 is resolved, the concerns regarding the validity of the motivation and the experimental scope remain. Properly addressing remaining concerns would require significant new experiments and revisions to the paper. As such, I believe the paper requires further development. I believe that the required changes warrant a new review cycle. Therefore, I will maintain my current score.
>
>
> [7] Bjorck, Johan, et al. "Gr00t n1: An open foundation model for generalist humanoid robots." arXiv preprint arXiv:2503.14734 (2025).
>
> [8] Zhou, Zhongyi, et al. "Vision-Language-Action Model with Open-World Embodied Reasoning from Pretrained Knowledge." arXiv preprint arXiv:2505.21906 (2025).

---

> ### Author Response · Authors · 2025-11-28
> **Response**
>
> We sincerely thank the reviewer for the thoughtful follow-up comments. However, we may disagree with several of the reviewer’s interpretations.
>
> W1: We firmly disagree with the reviewer’s claim. Our method **does not rely on classical weighted regression**, and this seems to be **a substantial misunderstanding**. As discussed in Appendix B.6, advantage weighting is not a regression heuristic but the theoretically derived variational objective for guiding **generative model–based policies**. Our goal is to **determine the most principled and effective training objective** for this broad class of ODE-based generative policies—not to restrict ourselves to a particular regression form. Moreover, as shown in Section 5.3 (Table 3), we explicitly evaluate direct Q-function differentiation: incorporating a linear Q-term indeed **achieves strong performance**, surpassing Diffusion-QL. The issue is therefore **not methodological incompatibility**, but rather that direct Q-gradients are extremely **sensitive to the choice of** λ, whose appropriate scale varies significantly across tasks due to inconsistent Q-value magnitudes. This makes such objectives brittle and difficult to tune in practice. For these reasons, we argue that the advantage-weighted objective provides a **more stable, efficient**, and generally applicable formulation for generative model–based policies, and is better aligned with the underlying continuous-time generative structure. We copy part of Table 3 below.
>
> | **Method**                                    | **Training Time** | **Score**          |
> |-----------------------------------------------|-------------------|--------------------|
> | **GTP (ours)**                                | 4.26 h            | **112.2 ± 0.6**    |
> | GTP-BC + linear Q-term (λ = 0.01)             | 5.08 h            | 111.4 ± 0.9        |
>
> W5: First, we must **explicitly correct the reviewer's misinterpretation** of our previous response: **we never stated we would not run the FQL experiments.** We clarified that we prioritized critical ablation studies given the limited rebuttal timeframe, not that we refused to include the baseline. We are implementing and running FQL currently. Second, we must clarify that **GTP is not merely a "parameterization strategy" (backbone choice).** As detailed in **Section 3.1**, standard Diffusion and Flow Matching (including FQL) primarily define _choices of vector fields_ that require expensive _iterative ODE solvers_. **Comparing FQL with GTP would primarily serve to provide a representative baseline for flow-matching policies that still rely on iterative ODE solvers.** However, this overlooks the core contribution: GTP proposes a **unified solver framework** that learns a **direct mapping** to the solution, effectively bypassing the iterative integration process entirely. Therefore, comparing against **Consistency-AC**—the state-of-the-art for _accelerated direct solving_—is the scientifically more appropriate benchmark, and our results already validate the superiority of our **direct mapping paradigm** over iterative approaches.

---

> > ### Author Response · Authors · 2025-11-28
> > **Response to W2,3**
> >
> > W2,3:  We still maintain that the efficiency bottleneck is a critical issue for standard diffusion.
> > 1. The **dependency on high sampling steps** is widely recognized as a fundamental characteristic of standard diffusion, validated across multiple domains. In **RL**, seminal works like **Diffuser** [1] require **20 or 100 steps** for effective planning, and the emergence of dedicated acceleration methods (e.g., **Consistency-AC** [2]) explicitly confirms that inference latency is a community-acknowledged bottleneck. This is not unique to RL. State-of-the-art Diffusion LLMs like **LLaDA** [3] typically require **32–64+ steps** to match autoregressive performance, highlighting that the generative mechanism inherently demands significant computation. This cross-domain evidence confirms that without specific algorithmic changes (like ours or Flow Matching), standard diffusion inherently demands significant computation for high-quality generation.
> > 2. Regarding the reviewer's observation that `Diffusion-QL` works at $T=5$ on `Hopper`: This reflects the simplicity of low-dimensional D4RL tasks where performance **saturates** quickly. However, **this observation does not generalize to complex settings**. As evidenced by the **N=100** requirement for manipulation tasks in [1], standard diffusion suffers from performance degradation on complex multimodal distributions when steps are aggressively reduced.
> > 3. We must point out a **significant inaccuracy** regarding the cited papers. The reviewer refers to "pi0.5" but cites **[4] Zhou et al.**, which corresponds to _ChatVLA-2_. The actual **pi0.5** model is developed by _Physical Intelligence_ [5]. Crucially, **both** the actual **pi0.5** [5] and **GR00T** [6] are explicitly **Flow Matching** models, _not_ standard diffusion. As stated in the pi0 methodology, the model _"adds action outputs that use **flow matching** to generate continuous action distributions"_ (similar for GR00T). They achieve few-step inference only by fundamentally changing the generative paradigm to "straighten" ODE trajectories. Furthermore, **both models** achieve this speed by shifting the computational burden to **massive pre-training**. **For pi0.5**, it explicitly states it is _"based on the **PaliGemma** vision-language model,"_ utilizing a **3-billion parameter** backbone with internet-scale priors. **GR00T** is explicitly defined as a **'Foundation Model'**, which **entails massive pre-training** on large-scale human and robot datasets to acquire generalist capabilities. The fact that these SOTA models _shifted_ to Flow Matching specifically to mitigate latency **validates our premise**: standard diffusion is indeed slow. Even so, **GTP** achieves high performance at **T=2**—**faster** than the ~10-step requirement of these massive Flow models—effectively resolving the bottleneck within the standard diffusion framework without requiring billions of parameters. Ultimately, given the community's clear imperative for real-time, high-performance control, we find it counterintuitive to dismiss a unified framework that successfully resolves this long-standing trade-off. Our work demonstrates that it is possible to achieve SOTA performance with extreme efficiency ($T=2$) _without_ the massive computational debt of foundation models, representing a significant and necessary advancement for the field.
> >
> > **References:**
> >
> > [1] Janner, Michael, et al. Planning with Diffusion for Flexible Behavior Synthesis. International Conference on Machine Learning. PMLR, 2022. \
> > [2] Zihan Ding and Chi Jin. Consistency models as a rich and efficient policy class for reinforcement learning. In International Conference on Learning Representations, 2024. \
> > [3] Nie, Shen, et al. Large Language Diffusion Models. ICLR 2025 Workshop on Deep Generative Model in Machine Learning: Theory, Principle and Efficacy. \
> > [4] Zhou, Zhongyi, et al. Vision-Language-Action Model with Open-World Embodied Reasoning from Pretrained Knowledge. arXiv preprint arXiv:2505.21906 (2025). \
> > [5] Intelligence, Physical, et al. $\pi_ {0.5} $: a Vision-Language-Action Model with Open-World Generalization. arXiv preprint arXiv:2504.16054 (2025). \
> > [6] Bjorck, Johan, et al. Gr00t n1: An open foundation model for generalist humanoid robots. arXiv preprint arXiv:2503.14734 (2025).

---

> ### Author Response · Authors · 2025-12-03
> **Supplementary experiments of Flow-Q Learning**
>
> We implemented Flow Q-Learning and included the latest code in the supplementary material. Regarding FQL, we would like to clarify several points:
> 1. FQL trains a flow-matching model for behavior cloning with $T=10$ and then distills a one-step policy from it. Therefore, it is essentially a distilled model and requires **more sampling steps** to obtain a sufficiently strong teacher model for distillation. Under this setup, a direct comparison may not be entirely fair, since FQL inherently depends on a high-step teacher model while our method does not.
> 2. FQL also uses an additive Q-loss, which makes the weighting coefficient $\alpha$ highly sensitive. This aligns with our earlier observation that additive Q-losses often require task-specific fine-tuning, which is why we avoid using them in our method.
> 3. Due to the limited rebuttal time and the need to re-tune FQL’s hyperparameters (the FQL paper does not provide settings for the Gym tasks), we only report the results we currently have. The remaining experiments will be included in the final version.
>
> Despite the remaining FQL experiments, this does not affect our main claim: our unified ODE-based paradigm remains general and fully compatible with flow-matching–based methods such as FQL.
>
>
>
> | Task                      | FQL                                           | GTP       |
> | ------------------------- | --------------------------------------------- | --------- |
> | hopper-medium-expert-v2   | 56.2 ± 4.6 ($\alpha=10$), $\alpha=1$ diverged | 112.2±0.6 |
> | antmaze-umaze-v0          | 100                                           | 100       |
> | antmaze-medium-diverse-v0 | 85.2±1.0                                      | 94.2±2.0  |

---

### Author Response · Authors · 2025-11-21
**General response**

Dear reviewers,

Thank you for your valuable feedback and recognizing the following strengths of our paper:
- **Theoretical contribution & unified framework.** It includes our continuous-time ODE solution-map formulation and the accompanying theoretical guarantees (e.g., Theorem 1). (all the reviewers)
- **SOTA results** across both behavior cloning and offline RL benchmarks. (all the reviewers)

Here, we addressee several possible concerns:
1. What is the novelty of this paper?
	1. We introduce a **unified continuous-time ODE solution-map framework** showing that CM/CTM/Shortcut/Mean Flows are all special cases differing only in how they approximate the average velocity. This perspective has not appeared before and directly leads to a **new generative policy class** tailored for RL.
	2. We further propose an **approximate score**—together with Theorem 1—that provides a practical and theoretically justified supervision signal for RL. Without this approximation, the unified ODE framework would be impractical for RL because ODE-solver-based supervision (as in CTMs) is prohibitively expensive.  This approximate score is what makes the unified framework **actually usable** inside large-scale training.
2. How is GTP different from CTM?
	1. GTP is **not** CTM adapted to RL. Our goal is to _identify the common structure_ across CM/CTM/Shortcut/MeanFlows and describe them in a unified way via average velocity. CTM, Shortcut Models, and Mean Flows can all be seen as reparameterizations of the same underlying ODE trajectory. We adopt a CTM-like parameterization because directly predicting $x_{s}$​ is efficient and aligns with the instantaneous-flow interpretation. Our consistency loss is also **simpler** and arises naturally from the ODE solution-map formulation.
	2. CTM requires multi-step ODE supervision (20–40 steps) and DSM/GAN auxiliaries; GTP uses **one-step supervision** via Theorem 1. This is the fundamental difference: GTP is much closer to **flow matching** than to CTM, despite having a superficially similar form. (See Appendix B.4 for detailed discussion.)
3. Is our value-guided loss just AWR?
	1. No. Our goal is to determine **how to perform policy improvement for a generative trajectory model** whose main supervision comes from behavior-cloning–style losses. Although adding a Q-term (as in Diffusion-QL or Consistency-AC) is possible, our model already uses two generative losses, and introducing a third term makes the weighting highly sensitive. This motivates using a cleaner alternative. We also compare our weighted loss with an additive Q-loss (Section 5.3, Table 3). After tuning its hyperparameter, the Q-loss baseline can achieve reasonable performance, but it remains more sensitive to task-specific tuning and, in practice, less efficient than our weighted loss.
	2. Our solution is distribution-level reweighting, which keeps training in-distribution and naturally prioritizes high-value samples. This yields an advantage-weighted objective that improves sample efficiency without introducing extra loss balancing. While advantage weighting is not new, our use of it is driven by the structural needs of our unified ODE framework rather than by borrowing prior heuristics.
4. What is the performance of GTP under $T=2$?
	1. Consistent with prior findings in Consistency-AC, GTP’s performance almost **saturates at** $T=2$. Increasing $T$ beyond 2 yields negligible additional gains but increases compute.
	2. The partial $T=2$ results we have completed show that GTP with $T=2$ achieves **slightly lower but very close performance** compared to $T=5$, while being substantially more efficient (0.67ms for one sample, see Table 6). We provide the available results below.


| Gym                          | CTM-AC (T=2) | CTM-AC (T=5)    | Diffusion-AC   | Consistency-AC |
| ---------------------------- | ------------ | --------------- | -------------- | -------------- |
| halfcheetah-medium-v2        | 53.1 ± 0.5   | 53.9 ± 0.1      | 51.1 ± 0.5     | **69.1 ± 0.7**     |
| hopper-medium-v2             | 87.8 ± 2.3   | 90.3 ± 2.7      | **90.5 ± 4.6**     | 80.7 ± 10.5    |
| walker2d-medium-v2           | **90.5 ± 0.5**   | 89.5 ± 0.6      | 87.0 ± 0.9     | 83.1 ± 0.3     |
| halfcheetah-medium-replay-v2 | 48.7 ± 0.2   | 50.8 ± 0.4      | 47.8 ± 0.3     | **58.7 ± 3.9**     |
| hopper-medium-replay-v2      | 101.6 ± 0.5  | **101.7 ± 0.3**     | 95.5 ± 1.5 | 99.7 ± 0.5     |
| walker2d-medium-replay-v2    | **94.3±1.4**     | 94.2 ± 0.3      | 79.5 ± 3.6     | 79.5 ± 3.6     |
| halfcheetah-medium-expert-v2 | **96.2±0.4**     | 93.8 ± 0.8      | 84.3 ± 4.1     | 84.3 ± 4.1     |
| hopper-medium-expert-v2      | 111.7±0.6    | 112.2 ± 0.6     | **96.8 ± 0.3** | 100.4 ± 3.5    |
| walker2d-medium-expert-v2    | **114.2±1.0**    | **114.2 ± 0.3** | 110.1 ± 0.3    | 110.4 ± 0.7    |
| **Average**                  | 88.7         | **89.0**        | 87.9           | 85.1           |

---

> ### Author Response · Authors · 2025-11-25
>
> We have addressed the reviewers’ feedback and incorporated the corresponding revisions to strengthen the manuscript.
> The main changes are summarized below, and all modifications are highlighted in blue in the updated version.
> - We revised Section 3 to clarify that the unified framework is our contribution and added a subsection discussing its connections to prior work.
> - We added an ablation study evaluating GTP with the $T=2$ setting (Appendix D.1).
> - We added an sensitivity analysis of GTP with $T$ and $\eta$. (Appendix D.2)
> - We improved the ablation in Appendix D.3 by including inference-time comparisons for GTP with $T=2$.

---

### Author Response · Authors · 2025-12-03
**Summary**

Below we provide a concise summary of the paper and the key clarifications made during the rebuttal process.
1. **Novelty & Contribution.** We emphasize that our main contribution is a **unified ODE-based generative framework** showing that diffusion models, consistency models, CTMs, shortcut models, flow matching, and mean flows all emerge as special cases of learning the ODE solution map $\Phi$. This establishes a **new policy paradigm (GTP)** rather than an incremental modification of existing models.
2. **Efficiency & Stability.** We demonstrate that our **score approximation (Theorem 1)** replaces expensive solver-based supervision with a closed-form surrogate, and provably changes the objective only by $O(h^p)$. This removes the primary computational bottleneck and eliminates the instability caused by self-generated on-trajectory supervision.
3. **Value-Guided Learning.** We adopt an **advantage-weighted objective** not only because it requires minimal task-specific tuning, but also because it is inherently well-suited for generative models: it reweights the original data distribution in a principled way, keeping training strictly **in-distribution**. This prevents the actor from drifting toward unsupported actions and thus avoids **critic divergence**, a common failure mode of direct Q-loss formulations.
4. **Strength of Empirical Evidence.** Across behavior cloning, offline RL, extensive ablations, and multi-goal visualization, GTP consistently matches or surpasses prior methods while achieving a better expressiveness–efficiency balance.
5. In our rebuttal, we have addressed all major reviewers’ concerns (baseline fairness, hyper-parameter sensitivity, stability, comparative evaluation, value-guided rationale). None of these issues compromises our main conceptual or empirical contributions.

---

### Meta-Review · Area_Chair_defy · 2026-01-06

**Summary:**

This paper proposes Generative Trajectory Policies (GTP), framing diffusion, flow-matching, and consistency models under a unified ODE perspective and applying this framework to offline RL with advantage-weighted training. Experiments show strong performance on D4RL benchmarks, particularly AntMaze.

Reviewers acknowledged the clear unifying perspective, solid technical presentation, and competitive empirical results. However, concerns were repeatedly raised about the degree of novelty beyond existing consistency/flow-based methods, the incremental nature of the algorithmic contributions, and whether the ODE unification yields fundamentally new insights or capabilities. Reviewers also questioned the limited experimental scope (e.g., reliance on D4RL), sensitivity to design choices, and the strength of comparisons needed to isolate the benefits of the proposed framework.

After considering the reviews, rebuttal, and discussion, I find that while the paper is technically sound, the contributions do not sufficiently exceed prior work to meet the bar for acceptance at this venue.

**Reviewer Concerns:**

Please see my summary.

**Reviewer Scores:**

It is difficult to say.  Overall, the authors provided some solid rebuttal, but it's a subjective judgement for the reviewer whether they would like to raise their score.

---

### Decision · Program_Chairs · 2026-01-26

Reject